# Conditional Generation Using Polynomial Expansions

**Grigorios G Chrysos**
EPFL, Switzerland
grigorios.chrysos@epfl.ch

**Markos Georgopoulos**
Imperial College London, UK
m.georgopoulos@imperial.ac.uk

**Yannis Panagakis**
University of Athens, GR
yannisp@di.uoa.gr

## Abstract

Generative modeling has evolved to a notable field of machine learning. Deep polynomial neural networks (PNNs) have demonstrated impressive results in unsupervised image generation, where the task is to map an input vector (i.e., noise) to a synthesized image. However, the success of PNNs has not been replicated in conditional generation tasks, such as super-resolution. Existing PNNs focus on single-variable polynomial expansions which do not fare well to two-variable inputs, i.e., the noise variable and the conditional variable. In this work, we introduce a general framework, called CoPE, that enables a polynomial expansion of two input variables and captures their auto- and cross-correlations. We exhibit how CoPE can be trivially augmented to accept an arbitrary number of input variables. CoPE is evaluated in five tasks (class-conditional generation, inverse problems, edges-to-image translation, image-to-image translation, attribute-guided generation) involving eight datasets. The thorough evaluation suggests that CoPE can be useful for tackling diverse conditional generation tasks. The source code of CoPE is available at https://github.com/grigorisg9gr/polynomial_nets_for_conditional_generation.

## 1 Introduction

Modelling high-dimensional distributions and generating samples from complex distributions are fundamental tasks in machine learning. Among prominent generative models, StyleGAN [Karras et al., 2019] has demonstrated unparalleled performance in unsupervised image generation. Its success can be attributed to the higher-order correlations of the input vector $z$ captured by the generator. As Chrysos et al. [2019] argue, StyleGAN[1] is best explained as a deep polynomial neural network (PNN). PNNs have demonstrated impressive generation results in faces, animals, cars [Karras et al., 2020b], paintings, medical images [Karras et al., 2020a]. Nevertheless, PNNs have yet to demonstrate similar performance in conditional generation tasks, such as super-resolution or image-to-image translation.

In contrast to unsupervised generators that require a single-variable input $z$, in conditional generation (at least) two inputs are required: i) one (or more) conditional variables $c$, e.g., a low-resolution image, and ii) a noise sample $z$. A trivial extension of PNNs for conditional generation would be to concatenate all the input variables into a fused variable. The fused variable is then the input to the single-variable polynomial expansion of PNNs. However, the concatenation reduces the flexibility of the model significantly. For instance, concatenating a noise vector and a vectorized low-resolution image results in sub-optimal super-resolution, since the spatial correlations of the input image are lost in the vectorization. Additionally, the concatenation of the vectorized conditional variable $c$ and

---

[1]This work focuses on the generator network; any reference to StyleGAN refers to its generator.

35th Conference on Neural Information Processing Systems (NeurIPS 2021).

$z$ leads to a huge number of parameters when we use a fully-connected layer as typically done in the input of StyleGAN, especially when $c$ depicts an image.

In this work, we introduce a framework, called *CoPE*, for conditional data generation. CoPE resorts to multivariate polynomials that capture the higher-order auto- and cross-correlations between the two input variables. By imposing a tailored structure in the higher-order correlations, we obtain an intuitive, recursive formulation for CoPE. The formulation enables different constraints to be applied to each variable and its associated parameters. In CoPE, different architectures can be defined simply by changing the recursive formulation. Our contributions can be summarized as follows:

- We introduce a framework, called CoPE, that expresses a high-order, multivariate polynomial for conditional data generation. We exhibit how CoPE can be applied on diverse conditional generation tasks.
- We derive two extensions to the core two-variable model: a) we augment the formulation to enable an arbitrary number of conditional input variables, b) we design different architectures that arise by changing the recursive formulation.
- CoPE is evaluated on *five different tasks* (class-conditional generation, inverse problems, edges-to-image translation, image-to-image translation, attribute-guided generation); overall *eight datasets* are used for the thorough evaluation.

The diverse experiments suggest that CoPE can be useful for a variety of conditional generation tasks, e.g., by defining task-specific recursive formulations. To facilitate the reproducibility, the source code is available at https://github.com/grigorisg9gr/polynomial_nets_for_conditional_generation.

## 2 Related work

Below, we review representative works in conditional generation and then we summarize the recent progress in multiplicative interactions (as low-order polynomial expansion).

### 2.1 Conditional generative models

The literature on conditional generation is vast. The majority of the references below focus on Generative Adversarial Networks (GANs) [Goodfellow et al., 2014] since GANs have demonstrated the most impressive results to date, however similar methods can be developed for other generative models, such as Variational Auto-encoders (VAEs) [Kingma and Welling, 2014]. Four groups of conditional generative models are identified below based on the type of conditional information.

The first group is the *class-conditional generation* [Miyato et al., 2018, Brock et al., 2019, Kaneko et al., 2019], where the data are divided into discrete categories, e.g., a cat or a ship. During training a class label is provided and the generator should synthesize a sample from that category. One popular method of including class-conditional information is through conditional normalization techniques [Dumoulin et al., 2017, De Vries et al., 2017]. An alternative way is to directly concatenate the class labels with the input noise; however, as Odena et al. [2017] observe, this model does not scale well to a hundred or a thousand classes.

The second group is *inverse problems* that have immense interest for both academic and commercial reasons [Sood et al., 2018, You et al., 2019]. The idea is to reconstruct a latent signal, when corrupted measurements are provided [Ongie et al., 2020]. Well-known inverse problems in imaging include super-resolution, deblurring, inpainting. Before the resurgence of deep neural networks, the problems were tackled with optimization-based techniques [Chan and Chen, 2006, Levin et al., 2009]. The recent progress in conditional generation has fostered the interest in inverse problems [Ledig et al., 2017, Pathak et al., 2016, Huang et al., 2017]. A significant challenge that is often overlooked in the literature [Ledig et al., 2017, Yu et al., 2018a] is that there are many possible latent signals for each measurement. For instance, given a low-resolution image, we can think of several high-resolution images that when down-sampled provide the same low-resolution image.

Another group is *image-to-image translation*. The idea is to map an image from one domain to an image to another domain, e.g., a day-time image to a night-time image. The influential work of Isola et al. [2017] has become the reference point for image-to-image translation. Applications in conditional pose generation [Ma et al., 2017, Siarohin et al., 2018], conditional video generation [Wang

Table 1: Comparison of polynomial neural networks (PNNs). Even though the architectures[1] of Karras et al. [2019], Chen et al. [2019], Park et al. [2019] were not posed as polynomial expansions, we believe that their success can be (partly) attributed to the polynomial expansion (please check sec. F for further information). Π-Net and StyleGAN are not designed for conditional data generation. In practice, learning complex distributions requires high-order polynomial expansions; this can be effectively achieved with products of polynomials as detailed in sec. 3.2. Only Π-Net and CoPE include such a formulation. The columns on discrete and continuous variable refer to the type of conditional variable the method was originally proposed on, e.g., sBN was only tried on class-conditional generation. Additionally, the only work that enables multiple conditional variables (and includes related experiments) is the proposed CoPE.

| Attributes of polynomial-like networks. | | | | |
|---|---|---|---|---|
| Model | products of polynomials | discrete cond.variable | continuous cond. variable | multiple cond. variables |
| Π-Net [Chrysos et al., 2020] | ✓ | ✗ | ✗ | ✗ |
| StyleGAN [Karras et al., 2019] | ✗ | ✗ | ✗ | ✗ |
| sBN [Chen et al., 2019] | ✗ | ✓ | ✗ | ✗ |
| SPADE [Park et al., 2019] | ✗ | ✗ | ✓ | ✗ |
| CoPE (ours) | ✓ | ✓ | ✓ | ✓ |

et al., 2018a] or generation from semantic labels [Wang et al., 2018b] have appeared. Despite the success, converting the mapping from one-to-one to one-to-many, i.e., having multiple plausible outputs for a single input image, has required some work [Zhu et al., 2017b, Huang et al., 2018]. A more dedicated discussion on diverse generation is deferred to sec. I.

The fourth group uses multiple conditional variables for generation, e.g., attribute-guided generation [Choi et al., 2018]. These methods include significant engineering (e.g., multiple discriminators [Xu et al., 2017], auxiliary losses). The influential InfoGAN [Chen et al., 2016] explicitly mentions that without additional losses the generator is 'free to ignore' the additional variables.

Each technique above is typically applied to a single group of conditional generation tasks, while our goal is to demonstrate that CoPE can be applied to different tasks from these groups.

## 2.2 Multiplicative interactions

Multiplicative connections have long been adopted in machine learning [Shin and Ghosh, 1991, Hochreiter and Schmidhuber, 1997, Bahdanau et al., 2015, Rendle, 2010]. The idea is to combine the inputs through elementwise products or other diagonal forms. Jayakumar et al. [2020] prove that second order multiplicative operators can represent a greater class of functions than classic feed-forward networks. Even though we capitalize on the theoretical argument, our framework can express any higher-order correlations while the framework of Jayakumar et al. [2020] is limited to second order interactions.

Higher-order correlations have been studied in the tensor-related literature [Kolda and Bader, 2009, Debals and De Lathauwer, 2017]. However, their adaptation in modern deep architectures has been slower. Π-Net [Chrysos et al., 2020] resorts to a high-order polynomial expansion for mapping the input $z$ to the output $x = G(z)$. Π-Net focuses on a single-variable polynomial expansion; an in-depth difference of our work with Π-Net can be found in sec. E in the supplementary. Two efforts on conditional generation which can be cast as polynomial expansions are SPADE [Park et al., 2019] and sBN [Chen et al., 2019]. SPADE can be interpreted as a single-variable polynomial expansion with respect to the conditional variable $c$. SPADE does not capture the higher-order cross-correlations between the input variables. Similarly, sBN can be interpreted as a polynomial expansion of the two variables for class-conditional generation. However, SPADE and sBN do not use the product of polynomial formulation, which enables high-order expansions without increasing the number of layers [Chrysos et al., 2020]. Importantly, SPADE and sBN are constructed for specific applications (i.e., semantic image generation and unsupervised/class-conditional generation respectively) and it remains unclear whether a PNN can effectively tackle a general-purpose conditional generation task.

## 3 Method

In the following paragraphs we introduce the two-variable polynomial expansion (sec. 3.1), while the detailed derivation, along with additional models are deferred to the supplementary (sec. B). The crucial technical details, including the stability of the polynomial, are developed in sec. 3.2. We emphasize that a multivariate polynomial can approximate any function [Stone, 1948, Nikol'skii, 2013], i.e., a multivariate polynomial is a universal approximator.

**Notation**:Tensors/matrices/vectors are symbolized by calligraphic/uppercase/lowercase boldface letters e.g., $\mathcal{W},\boldsymbol{W},\boldsymbol{w}$. The *mode-$m$ vector product* of $\mathcal{W}$ (of order $M$) with a vector $\boldsymbol{u} \in \mathbb{R}^{I_m}$ is $\mathcal{W} \times_m \boldsymbol{u}$ and results in a tensor of order $M-1$. We assume that $\prod_{i=a}^{b} x_i = 1$ when $a > b$. The core symbols are summarized in Table 2, while a detailed tensor notation is deferred to the supplementary (sec. B.1).

Table 2: Symbols

| Symbol | Role |
|---|---|
| $N$ | Expansion order of the polynomial |
| $k$ | Rank of the decompositions |
| $\boldsymbol{z}_\mathrm{I}, \boldsymbol{z}_\mathrm{II}$ | Inputs to the polynomial |
| $n, \rho$ | Auxiliary variables |
| $\mathcal{W}^{[n,\rho]}$ | Parameter tensor of the polynomial |
| $\boldsymbol{U}_{[n]}, \boldsymbol{C}, \boldsymbol{\beta}$ | Learnable parameters |
| $*$ | Hadamard product |

### 3.1 Two input variables

Given two input variables [2] $\boldsymbol{z}_\mathrm{I}, \boldsymbol{z}_\mathrm{II} \in \mathbb{K}^d$ where $\mathbb{K} \subseteq \mathbb{R}$ or $\mathbb{K} \subseteq \mathbb{N}$, the goal is to learn a function $G : \mathbb{K}^{d \times d} \to \mathbb{R}^o$ that captures the higher-order correlations between the elements of the two inputs. We can learn such higher-order correlations as polynomials of two input variables. A polynomial expansion of order $N \in \mathbb{N}$ with output $\boldsymbol{x} \in \mathbb{R}^o$ (such that $\boldsymbol{x} = G(\boldsymbol{z}_\mathrm{I}, \boldsymbol{z}_\mathrm{II})$) has the form:

$$\boldsymbol{x} = \sum_{n=1}^{N} \sum_{\rho=1}^{n+1} \left( \mathcal{W}^{[n,\rho]} \prod_{j=2}^{\rho} \times_j \boldsymbol{z}_\mathrm{I} \prod_{\tau=\rho+1}^{n+1} \times_\tau \boldsymbol{z}_\mathrm{II} \right) + \boldsymbol{\beta} \quad (1)$$

where $\boldsymbol{\beta} \in \mathbb{R}^o$ and $\mathcal{W}^{[n,\rho]} \in \mathbb{R}^{o \times \prod_{m=1}^{n} \times_m d}$ for $n \in [1, N], \rho \in [1, n+1]$ are the learnable parameters. The expansion depends on two (independent) variables, hence we use the $n$ and $\rho$ as auxiliary variables. The two products of (1) do not overlap, i.e., the first multiplies the modes $[2, \rho]$ (of $\mathcal{W}^{[n,\rho]}$) with $\boldsymbol{z}_\mathrm{I}$ and the other multiplies the modes $[\rho+1, n+1]$ with $\boldsymbol{z}_\mathrm{II}$.

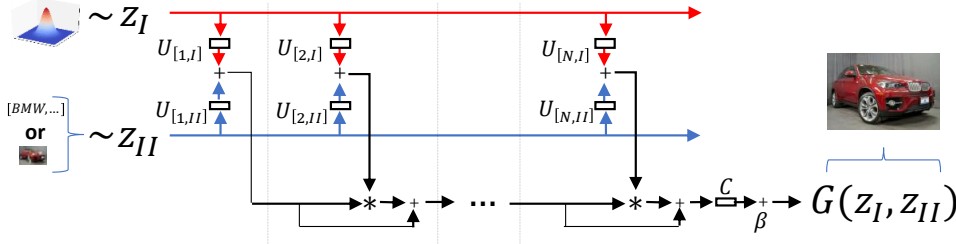

Figure 1: Abstract schematic for $N^{\text{th}}$ order approximation of $\boldsymbol{x} = G(\boldsymbol{z}_I, \boldsymbol{z}_{II})$. The inputs $\boldsymbol{z}_I, \boldsymbol{z}_{II}$ are symmetric in our formulation. We denote with $\boldsymbol{z}_I$ a noise vector, e.g., a samples from Gaussian distribution, while $\boldsymbol{z}_{II}$ symbolizes a sample from a conditional input (e.g., a class label or a low-resolution image).

**Recursive relationship**: The aforementioned derivation can be generalized to an arbitrary expansion order. The recursive formula for an arbitrary order $N \in \mathbb{N}$ is the following:

$$\boldsymbol{x}_n = \boldsymbol{x}_{n-1} + \left( \boldsymbol{U}_{[n,I]}^T \boldsymbol{z}_\mathrm{I} + \boldsymbol{U}_{[n,II]}^T \boldsymbol{z}_\mathrm{II} \right) * \boldsymbol{x}_{n-1} \quad (2)$$

---

[2]To avoid cluttering the notation we use same dimensionality for the two inputs. However, the derivations apply for different dimensionalities, only the dimensionality of the tensors change slightly.

for $n = 2, \ldots, N$ with $\boldsymbol{x}_1 = \boldsymbol{U}_{[1,I]}^T \boldsymbol{z}_{\text{I}} + \boldsymbol{U}_{[1,II]}^T \boldsymbol{z}_{\text{II}}$ and $\boldsymbol{x} = \boldsymbol{C}\boldsymbol{x}_N + \boldsymbol{\beta}$. The parameters $\boldsymbol{C} \in \mathbb{R}^{o \times k}, \boldsymbol{U}_{[n,\phi]} \in \mathbb{R}^{d \times k}$ for $n = 1, \ldots, N$ and $\phi = \{I, II\}$ are learnable.

The intuition behind this model is the following: An embedding is initially found for each of the two input variables, then the two embeddings are added together and they are multiplied elementwise with the previous approximation. The different embeddings for each of the input variables allows us to implement $\boldsymbol{U}_{[n,I]}$ and $\boldsymbol{U}_{[n,II]}$ with different constraints, e.g., $\boldsymbol{U}_{[n,I]}$ to be a dense layer and $\boldsymbol{U}_{[n,II]}$ to be a convolution.

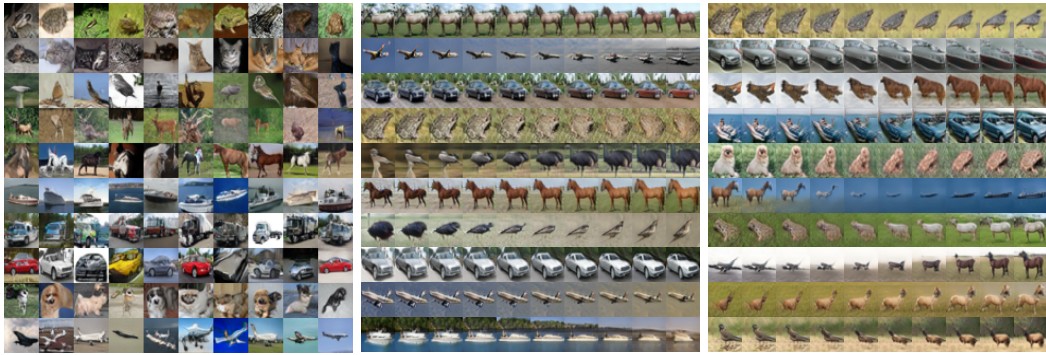

(a) Random samples per class      (b) Intra-class interpolation      (c) Inter-class interpolation

Figure 2: Synthesized images by CoPE in the class-conditional CIFAR10 (with resnet-based generator): (a) Random samples where each row depicts the same class, (b) Intra-class linear interpolation from a source to the target, (c) inter-class linear interpolation. In inter-class interpolation, the class labels of the leftmost and rightmost images are one-hot vectors, while the rest are interpolated in-between; the resulting images are visualized. In all three cases, CoPE synthesizes realistic images.

## 3.2 Model extensions and technical details

There are three limitations in (2). Those are the following: a) (2) describes a polynomial expansion of a two-variable input, b) each expansion order requires additional layers, c) high-order polynomials might suffer from unbounded values. Those limitations are addressed below.

Our model can be readily extended beyond two-variable input; an extension with three-variable input is developed in sec. C. The pattern (for each order) is similar to the two-variable input: a) a different embedding is found for each input variable, b) the embeddings are added together, c) the result is multiplied elementwise with the representation of the previous order.

The polynomial expansion of (2) requires $\Theta(N)$ layers for an $N^{\text{th}}$ order expansion. That is, each new order $n$ of expansion requires new parameters $\boldsymbol{U}_{[n,I]}$ and $\boldsymbol{U}_{[n,II]}$. However, the order of expansion can be increased without increasing the parameters substantially. To that end, we can capitalize on the product of polynomials. Specifically, let $N_1$ be the order of expansion of the first polynomial. The output of the first polynomial is fed into a second polynomial, which has expansion order of $N_2$. Then, the output of the second polynomial will have an expansion order of $N_1 \cdot N_2$. The second polynomial of degree $N_2$ can either be a polynomial of one variable (i.e., the output of the previous polynomial) or more variables (i.e., the output of the previous polynomial and one or more of the inputs). In both cases the total degree of the output of the second polynomial will be $N_1 \cdot N_2$. The choice of the type of polynomial in each case is a design option. The product of polynomials can be used with arbitrary number of polynomials; it suffices the output of the $\tau^{\text{th}}$ polynomial to be the input to the $(\tau + 1)^{\text{th}}$ polynomial. For instance, if we assume a product of $\Phi \in \mathbb{N}$ polynomials, where each polynomial has an expansion order of two, then the polynomial expansion is of $2^{\Phi}$ order. In other words, we need $\Theta(\log_2(N))$ layers to achieve an $N^{\text{th}}$ order expansion.

In algebra, higher-order polynomials are unbounded and can thus suffer from instability for large values. To avoid such instability, we take the following three steps: a) CoPE samples the noise vector from the uniform distribution, i.e., from the bounded interval of $[-1, 1]$, b) a hyperbolic tangent is used in the output of the generator as a normalization, i.e., it constrains the outputs in the bounded interval of $[-1, 1]$, c) batch normalization [Ioffe and Szegedy, 2015] is used to convert

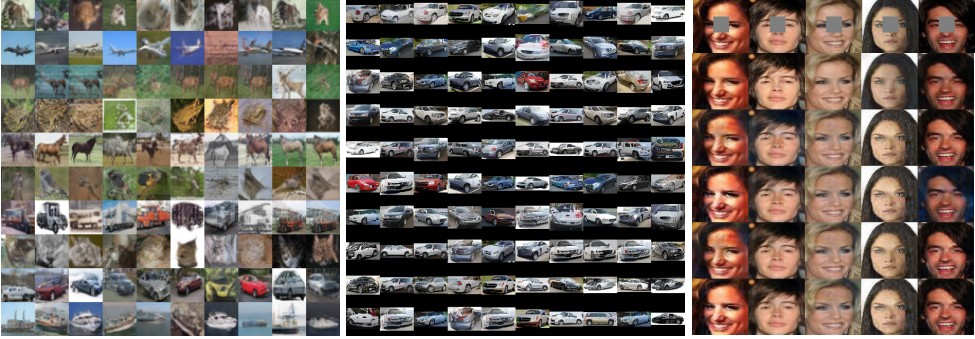

(a) Class-conditional generation    (b) Class-conditional generation    (c) Block-inpainting

Figure 3: Synthesized images by CoPE in the (a), (b) class-conditional generation (sec. 4.1) and (c) block-inpainting (sec. 4.3). In class-conditional generation, each row depicts a single class.

the representations to zero-mean. We emphasize that in GANs the hyperbolic tangent is the default activation function in the output of the generator, hence it is not an additional requirement of our method. Additionally, in our preliminary experiments, the uniform distribution can be changed for a Gaussian distribution without any instability. A theoretical analysis on the bounds of such multivariate polynomials would be an interesting subject for future work.

Lastly, we highlight the flexibility of the proposed CoPE. sBN and SPADE can be considered as special cases of the two-variable polynomial expansion. In particular, we exhibit in sec. F.1 how SPADE can be extended into a general-purpose two-variable polynomial expansion. In addition, the products of polynomials would enable both sBN and SPADE to perform higher-order expansions without increasing the number of layers.

## 4 Experiments

To validate the proposed formulation, the following diverse conditional generation tasks are considered:

- class-conditional generation trained on CIFAR10, Cars196 and SVHN in sec. 4.1 and sec. H.2.
- generation of unseen attribute combinations in sec. 4.2.
- attribute-guided generation in sec. H.5.
- inverse problems in imaging, e.g., super-resolution and block-inpainting, trained on Cars196 and CelebA in sec. 4.3.
- edges-to-image translation trained on handbags and shoes in sec. H.4.
- image-to-image translation in sec. H.3.

The details on the datasets and the evaluation metrics are deferred to the supplementary (sec. G) along with additional visualizations and experiments. Additionally, the source code of CoPE is available at https://github.com/grigorisg9gr/polynomial_nets_for_conditional_generation.

Our framework, e.g., (2), does not include any activation functions. To verify the expressivity of our framework, we maintain the same setting for the majority of the experiments below. Particularly, the generator does not have activation functions between the layers; there is only a hyperbolic tangent in the output space for normalization as typically done in GAN generators. However, we conduct one experiment using a strong baseline with activation functions. That is, a comparison with SNGAN [Miyato and Koyama, 2018] in class-conditional generation is performed (sec. 4.1).

**Baselines:** 'Π-Net-SICONC' implements a polynomial expansion of a single variable by concatenating all the input variables. 'SPADE' implements a polynomial expansion with respect to the conditional variable. Also, 'GAN-CONC' and 'GAN-ADD' are added as baselines, where we replace the Hadamard products with concatenation and addition respectively. A schematic of the differences between the compared polynomial methods is depicted in Fig. 7, while a detailed description of all methods is deferred to sec. G. Each experiment is conducted **five** times and the mean and the

Table 3: Quantitative evaluation on class-conditional generation with resnet-based generator (i.e., SNGAN). Higher Inception Score (IS) [Salimans et al., 2016] (lower Frechet Inception Distance (FID) [Heusel et al., 2017]) indicates better performance. The baselines improve the IS of SNGAN, however they cannot improve the FID. Nevertheless, SNGAN-CoPE improves upon all the baselines in both the IS and the FID.

| class-conditional generation on CIFAR10 | | |
|---|---|---|
| Model | IS ($\uparrow$) | FID ($\downarrow$) |
| SNGAN | $8.30 \pm 0.11$ | $14.70 \pm 0.97$ |
| SNGAN-CONC | $8.50 \pm 0.49$ | $30.65 \pm 3.55$ |
| SNGAN-ADD | $8.65 \pm 0.11$ | $15.47 \pm 0.74$ |
| SNGAN-SPADE | $8.69 \pm 0.19$ | $21.74 \pm 0.73$ |
| SNGAN-CoPE | $\mathbf{8.77 \pm 0.12}$ | $\mathbf{14.22 \pm 0.66}$ |

Table 4: Quantitative evaluation on class-conditional generation with $\Pi$-Net-based generator. In CIFAR10, there is a considerable improvement on the IS, while in Cars196 FID drops dramatically with CoPE. We hypothesize that the dramatic improvement in Cars196 arises because of the correlations of the classes. For instance, the SUV cars (of different carmakers) share several patterns, which are captured by our high-order interactions, while they might be missed when learning different normalization statistics per class. The generator *does not* have activation functions between the layers, so the deteriorated performance of GAN-CONC and GAN-ADD is reasonable.

| class-conditional generation on CIFAR10 | | | | class-conditional generation on Cars196 | |
|---|---|---|---|---|---|
| Model | IS ($\uparrow$) | FID ($\downarrow$) | | Model | FID ($\downarrow$) |
| GAN-CONC | $3.73 \pm 0.32$ | $294.33 \pm 8.16$ | | GAN-CONC | $240.45 \pm 16.79$ |
| GAN-ADD | $3.74 \pm 0.60$ | $298.53 \pm 16.54$ | | GAN-ADD | $208.72 \pm 12.65$ |
| SPADE | $4.00 \pm 0.53$ | $294.21 \pm 16.33$ | | SPADE | $168.19 \pm 39.71$ |
| $\Pi$-Net-SICONC | $6.65 \pm 0.60$ | $71.81 \pm 33.00$ | | $\Pi$-Net-SICONC | $153.39 \pm 27.93$ |
| $\Pi$-Net | $7.54 \pm 0.16$ | $37.26 \pm 1.86$ | | $\Pi$-Net | $120.40 \pm 28.65$ |
| CoPE | $\mathbf{7.87 \pm 0.21}$ | $\mathbf{34.35 \pm 2.68}$ | | CoPE | $\mathbf{55.48 \pm 3.16}$ |

standard deviation are reported. Throughout the experimental section, we reserve the symbol $z_{\Pi}$ for the conditional input (e.g., a class label).

## 4.1 Class-conditional generation

In class-conditional generation the conditional input is a class label in the form of one-hot vector. The experiments we conduct below modify only the generator, while in all cases we assume there is the same discriminator. In particular, two types of generators are used: a) a resnet-based generator (SNGAN), b) a polynomial generator ($\Pi$-Net). The former network has exhibited strong performance the last few years, while the latter is a recently proposed PNN.

**Resnet-based generator:** The experiment is conducted by augmenting the resnet-based generator of SNGAN[3]. Each compared method will be named according to the modification on the generator of SNGAN, e.g., SNGAN-CoPE utilizes a resnet-based generator following the proposed framework of (14). All methods are trained using CIFAR10 images; CIFAR10 is a popular benchmark in class-conditional generation. The quantitative results are in Table 3 and synthesized samples are illustrated in Fig. 2(a). SNGAN-CoPE improves upon all the baselines in both the Inception score (IS) [Salimans et al., 2016] and the FID [Heusel et al., 2017]. The proposed formulation enables inter-class interpolations. That is, the noise $z_{\mathrm{I}}$ is fixed, while the class $z_{\Pi}$ is interpolated. In Fig. 2(b) and Fig. 2(c), intra-class and inter-class linear interpolations are illustrated respectively. Both the quantitative and the qualitative results exhibit the effectiveness of our framework.

**$\Pi$-Net-based generator:** A polynomial expansion is selected as the baseline architecture for the generator[3]. In the original $\Pi$-Net conditional batch normalization (CBN) was used in the generator; this is replaced by batch normalization in the rest of the compared methods. The quantitative results in CIFAR10 are summarized in Table 4 (left). SPADE does not utilize the products of polynomials

---

[3]Further implementation details are offered in sec. G.

**FN**  **FS**  **MN**  **MS**

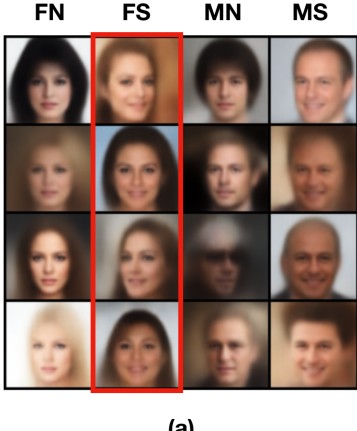 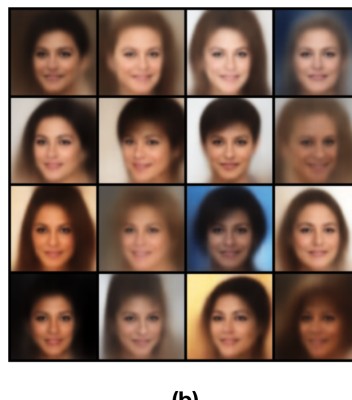

(a) (b)

Figure 4: Synthesized images with CoPE-VAE. In (a), all combinations are illustrated (the red is the combination missing during training, i.e. Female+Smile), while in (b), only images from the missing combination are visualized.

formulation, which explains its poor performance. Additionally, even though Π-Net-SINCONC and Π-Net both express a single-variable polynomial expansion, the inductive bias inserted into the network has a substantial effect in the final performance. Notice that CoPE outperforms all the baselines by a large margin.

We also evaluate class-conditional generation in Cars196 that has 196 classes. Cars196 is selected as a reasonably larger dataset than CIFAR10; it contains 196 classes and yet it can be trained on a single GPU. The compared methods and the training details remain the same. The results in Table 4 (right) demonstrate a substantial difference between CoPE and the compared methods. Namely, the proposed method achieves a 53.9% reduction of the FID over the best-performing baseline. We emphasize that both SPADE and Π-Net were not originally built for class-conditional generation, however we have tried to optimize the respective hyper-parameters to optimize their performance. The performance gap between SPADE and the rest PNNs can be explained by the lack of products of polynomials. We also verify the experimental results on CIFAR10 that demonstrate how Π-Net improves upon Π-Net-SINCONC. However, even Π-Net obtains a substantially higher FID than CoPE; we hypothesize that the improvement arises because of the correlations between the classes. For instance, the SUV cars of different carmakers share several patterns. Such correlations are captured by our framework, while they might be missed when learning different normalization statistics per class. Overall, CoPE synthesizes plausible images (Fig. 3) even in the absence of activation functionsn.

### 4.2 Polynomial conditioning for generating unseen attribute combinations

The proposed CoPE is a general framework for conditional generation, and we have already demonstrated how it can be used for conditional generation in GANs. In this section, we extend our method to the Variational Autoencoder (VAE) [Kingma and Welling, 2014] to showcase a useful byproduct of our formulation, namely the ability to generate unseen label combinations. In particular, as introduced in [Georgopoulos et al., 2020], this is a multi-label setting where one (or more) combinations are not seen in the training set. The method is then evaluated based on its ability to generate the unseen attribute combinations. To this end, we implement CoPE-VAE, a variation of the conditional VAE [Kingma and Welling, 2014] where both the encoder and decoder are polynomials. We perform experiments on CelebA using the annotated attributes of smile and gender. Similar to [Georgopoulos et al., 2020] we remove the combination (Smiling, Female) from the training set. The results in Figure 4 highlight the efficacy of the proposed conditioning method in disentangling the two labels and leveraging the multiplicative interactions to synthesize the missing combination.

### 4.3 Inverse problems in imaging

In the following paragraphs we evaluate the performance of CoPE in inverse problems. We select super-resolution and block-inpainting as two popular tasks.

Table 5: Quantitative evaluation on super-resolution with Π-Net-based generator on Cars196. The task on the left is super-resolution $16\times$, while on the right the task is super-resolution $8\times$. Our variant of SPADE, i.e., SPADE-CoPE (details in sec. G), vastly improves the original SPADE. The full two-variable model, i.e., CoPE, outperforms the compared methods.

| Super-resolution $16\times$ Cars196 | | Super-resolution $8\times$ Cars196 | | | |
|---|---|---|---|---|---|
| Model | FID ($\downarrow$) | Model | FID ($\downarrow$) | SSIM ($\uparrow$) | LPIPS ($\downarrow$) |
| SPADE | $111.75 \pm 13.41$ | SPADE | $119.18 \pm 14.82$ | 0.32 | 0.178 |
| Π-Net-SICONC | $80.16 \pm 12.42$ | Π-Net-SICONC | $186.42 \pm 40.84$ | 0.31 | 0.200 |
| SPADE-CoPE | $72.63 \pm 3.18$ | SPADE-CoPE | $64.76 \pm 8.26$ | 0.49 | 0.135 |
| CoPE | $\mathbf{60.42 \pm 6.19}$ | CoPE | $\mathbf{62.76 \pm 4.37}$ | **0.53** | **0.127** |

The core architectures remain as in the experiment above, i.e., CoPE and Π-Net-SICONC implement products of polynomials. A single change is made in the structure of the discriminator: Motivated by [Miyato and Koyama, 2018], we include an elementwise product of $z_{\mathrm{II}}$ with the real/fake image in the discriminator. This stabilizes the training and improves the results. Even though architectures specialized for a single task (e.g., Ledig et al. [2017]) perform well in that task, their well-selected inductive biases (e.g., perceptual or $\ell_1$ loss) do not generalize well in other domains or different conditional inputs. Our goal is not to demonstrate state-of-the-art results, but rather to scrutinize the effectiveness of the proposed formulation in different conditional generation tasks. To that end, we consider Π-Net-SICONC, SPADE and SPADE-CoPE as the baselines.

We experiment with two settings in super-resolution: one that the input image is down-sampled $8\times$ and one that it is down-sampled $16\times$. The two settings enable us to test the granularity of CoPE at different scales. In super-resolution $16\times$, $z_{\mathrm{II}}$ (i.e., the low-resolution input) has $48$ dimensions, while in super-resolution $8\times$, $z_{\mathrm{II}}$ has $192$ dimensions. The FID scores in Cars196 for the task of super-resolution are reported in Table 5. In addition, for the experiment on super-resolution $8\times$, the SSIM [Wang et al., 2004] and LPIPS [Zhang et al., 2018] are reported as widely-used metrics in inverse imaging tasks. Notice that the performance of Π-Net-SICONC deteriorates substantially when the dimensionality of the conditional variable increases. That validates our intuition about the concatenation in the input of the generator (sec. E), i.e., that the inductive bias of single-variable PNNs might not fare well in conditional generation tasks. We also report the SPADE-CoPE, which captures higher-order correlations with respect to the first variable as well (further details in sec. G). The proposed SPADE-CoPE outperforms the original SPADE, however it cannot outperform the full two-variable model, i.e., CoPE. The results indicate that CoPE performs well even when the conditional input is an image.

Beyond the quantitative results, qualitative results provide a different perspective on what the mappings learn. Qualitative results on the super-resolution experiments on Cars196 are provided in Fig. 8. We also provide synthesized results on both super-resolution $8\times$ and block-inpainting on CelebA in Fig. 8 and Fig. 3 respectively. For each conditional image, different noise vectors $z_{\mathrm{I}}$ are sampled. Notice that the corresponding synthesized images differ in the fine details. For instance, changes in the mouth region, the car type or position and even background changes are observed. Thus, CoPE synthesizes realistic images that i) correspond to the conditional input, ii) vary in the fine details. Similar variation has emerged even when the source and the target domains differ substantially, e.g., in the translation of MNIST digits to SVHN digits (sec. H.3). We should mention that the aforementioned experiments were conducted only using the adversarial learning loss. In the literature, regularization techniques have been proposed specifically for image-to-image translation, e.g., Yang et al. [2019], Lee et al. [2019]. However, such works utilize additional losses and even require additional networks for training, which makes the training computationally demanding and more sensitive to design choices.

## 5   Discussion

CoPE can be used for various conditional generation tasks as the experimental evaluation in both sec. 4 and the supplementary illustrate. Namely, CoPE can synthesize diverse content, which is typically tackled using auxiliary losses or networks in conditional GANs. We expect this attribute of diverse generation to be useful in inverse tasks, where multiple latent sharp images can correspond to a single corrupted image. The diverse generation can be attributed to the higher-order correlations between

the noise and the conditional variable. Such higher-order correlations also enable synthesizing images with unseen attribute combinations (sec. 4.2).

One limitation of our work is that our method has not been tried in large-scale synthesis, e.g., like Brock et al. [2019]. However, only a very limited number of labs/institutions have access to such resources. In addition, our single-GPU training does have a reduced energy footprint when compared to the multi-GPU setups of large scale GANs. We believe that the proposed method has merit despite the single-GPU training. Indeed, we demonstrate that CoPE can perform well in different tasks, which could help reduce the search for independent methods for every single task.

An interesting future step would be to evaluate the performance of the proposed CoPE in training with limited data, e.g., using techniques similar to Karras et al. [2020a]. This task can also encourage architecture discovery, similar to the various architectures devised in Chrysos et al. [2020]. In this work, we have demonstrated two such architectures with recursive formulations as in (2) and (14), however additional architectures can be designed by changing the tensor decomposition.

The equations of (2) and (14) express polynomial expansions of arbitrary order, which can approximate the target function without using activation functions between the recursive terms. However, in sec. 4.1 we have also experimented with generators that include activation functions between recursive terms (i.e., the experiment with SNGAN variants). In the future, we intend to study how to include the activation functions in the formulation and study the properties of such piecewise polynomial expansions. Lastly, demystifying the relationship between the order of the polynomial expansion and the expressivity (e.g., the implementation details for the order in sec. G) is a promising direction.

**Societal impact of image generation**: Manipulation of images is made possible through algorithms and architectures like ours with well-studied potential negative applications. The rapid progress in GAN-based image synthesis has made the discussion imperative. We encourage further work to be conducted on understanding how to detect synthesized images, e.g., in the conditional generation setting. For instance, our method can be used for training powerful classifiers that detect synthesized images.

## 6   Conclusion

We have introduced CoPE for conditional data generation. CoPE expresses a polynomial expansion of two input variables, i.e., a noise vector and a conditional variable. We exhibit how previously published methods, such as SPADE and sBN, can be considered as special forms of this two-variable polynomial expansion. Notably, CoPE can be augmented to accept an arbitrary number of conditional variables as inputs. The empirical evaluation confirms that our framework can synthesize realistic images in five diverse tasks, including inverse problems and class-conditional generation. Inverse problems, such as super-resolution, can benefit from the proposed framework; we showcase that sampling different noise vectors results in plausible differences in the synthesized image. We derive two recursive formulations, i.e., (2) and (14), but a new task-specific formulation can be easily defined. We expect this to be useful in learning from different modalities, such as visual question answering (VQA) or text-to-speech synthesis, since CoPE can capture high-order auto- and cross-correlations among the input variables.

## Acknowledgements

This project was sponsored by the Department of the Navy, Office of Naval Research(ONR) under a grant number N62909-17-1-2111. We are thankful to the reviewers for their constructive feedback and recommendations on how to improve our manuscript.

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
