# A  Summary of sections in the Appendix

In the following sections, further details and derivations are provided to elaborate the details of the CoPE. Specifically, in sec. B the decomposition and related details on the method are developed. The extension of our method beyond two-input variables is studied in sec. C. A method frequently used in the literature for fusing information is concatenation; we analyze how concatenation captures only additive and not more complex correlations (e.g., multiplicative) in sec. D. The differences from $\Pi$-Net [Chrysos et al., 2020] is explored in sec. E. In sec. F, some recent (conditional) data generation methods are cast into the polynomial neural network framework and their differences from the proposed framework are analyzed. The experimental details including the evaluation metrics and details on the baselines are developed in sec. G. In sec. H, additional experimental results are included. Lastly, the differences from works that perform diverse generation are explored in sec. I.

# B  Method derivations

In this section, we expand on the method details, including the scalar output case or the notation. Specifically, a more detailed notation is determined in sec. B.1; the scalar output case is analyzed in sec. B.2. In sec. B.3 a second order expansion is assumed to illustrate the connection between the polynomial expansion and the recursive formula. Sequentially, we derive an *alternative* model with different factor sharing. This model, called Nested-CoPE, has a nested factor sharing format (sec. B.4).

## B.1  Notation

Our derivations rely on tensors (i.e., multidimensional equivalent of matrices) and (tensor) products. We relay below the core notation used in our work, the interested reader can find further information in the tensor-related literature [Kolda and Bader, 2009, Debals and De Lathauwer, 2017].

**Symbols of variables**: Tensors/matrices/vectors are symbolized by calligraphic/uppercase/lowercase boldface letters e.g., $\boldsymbol{\mathcal{W}}, \boldsymbol{W}, \boldsymbol{w}$.

**Matrix products**: The *Hadamard* product of $\boldsymbol{A}, \boldsymbol{B} \in \mathbb{R}^{I \times N}$ is defined as $\boldsymbol{A} * \boldsymbol{B}$ and is equal to $a_{(i,j)} b_{(i,j)}$ for the $(i,j)$ element. The *Khatri-Rao* product of matrices $\boldsymbol{A} \in \mathbb{R}^{I \times N}$ and $\boldsymbol{B} \in \mathbb{R}^{J \times N}$ is denoted by $\boldsymbol{A} \odot \boldsymbol{B}$ and yields a matrix of dimensions $(IJ) \times N$. The Khatri-Rao product for a set of matrices $\{\boldsymbol{A}_{[m]} \in \mathbb{R}^{I_m \times N}\}_{m=1}^{M}$ is abbreviated by $\boldsymbol{A}_{[1]} \odot \boldsymbol{A}_{[2]} \odot \cdots \odot \boldsymbol{A}_{[M]} \doteq \bigodot_{m=1}^{M} \boldsymbol{A}_{[m]}$.

**Tensors**: Each element of an $M^{th}$ order tensor $\boldsymbol{\mathcal{W}}$ is addressed by $M$ indices, i.e., $(\boldsymbol{\mathcal{W}})_{i_1, i_2, \ldots, i_M} \doteq w_{i_1, i_2, \ldots, i_M}$. An $M^{th}$-order tensor $\boldsymbol{\mathcal{W}}$ is defined over the tensor space $\mathbb{R}^{I_1 \times I_2 \times \cdots \times I_M}$, where $I_m \in \mathbb{Z}$ for $m = 1, 2, \ldots, M$. The *mode-$m$ unfolding* of a tensor $\boldsymbol{\mathcal{W}} \in \mathbb{R}^{I_1 \times I_2 \times \cdots \times I_M}$ maps $\boldsymbol{\mathcal{W}}$ to a matrix $\boldsymbol{W}_{(m)} \in \mathbb{R}^{I_m \times \bar{I}_m}$ with $\bar{I}_m = \prod_{\substack{k=1 \\ k \neq m}}^{M} I_k$ such that the tensor element $w_{i_1, i_2, \ldots, i_M}$ is mapped to the matrix element $w_{i_m, j}$ where $j = 1 + \sum_{\substack{k=1 \\ k \neq m}}^{M} (i_k - 1) J_k$ with $J_k = \prod_{\substack{n=1 \\ n \neq m}}^{k-1} I_n$. The *mode-$m$ vector product* of $\boldsymbol{\mathcal{W}}$ with a vector $\boldsymbol{u} \in \mathbb{R}^{I_m}$, denoted by $\boldsymbol{\mathcal{W}} \times_m \boldsymbol{u} \in \mathbb{R}^{I_1 \times I_2 \times \cdots \times I_{m-1} \times I_{m+1} \times \cdots \times I_M}$, results in a tensor of order $M - 1$:

$$(\boldsymbol{\mathcal{W}} \times_m \boldsymbol{u})_{i_1, \ldots, i_{m-1}, i_{m+1}, \ldots, i_M} = \sum_{i_m=1}^{I_m} w_{i_1, i_2, \ldots, i_M} u_{i_m}. \tag{3}$$

We denote $\boldsymbol{\mathcal{W}} \times_1 \boldsymbol{u}^{(1)} \times_2 \boldsymbol{u}^{(2)} \times_3 \cdots \times_M \boldsymbol{u}^{(M)} \doteq \boldsymbol{\mathcal{W}} \prod_{m=1}^{m} \times_m \boldsymbol{u}^{(m)}$.

The *CP decomposition* [Kolda and Bader, 2009] factorizes a tensor into a sum of component rank-one tensors. The rank-$R$ CP decomposition of an $M^{th}$-order tensor $\boldsymbol{\mathcal{W}}$ is written as:

$$\boldsymbol{\mathcal{W}} \doteq [\![\boldsymbol{U}_{[1]}, \boldsymbol{U}_{[2]}, \ldots, \boldsymbol{U}_{[M]}]\!] = \sum_{r=1}^{R} \boldsymbol{u}_r^{(1)} \circ \boldsymbol{u}_r^{(2)} \circ \cdots \circ \boldsymbol{u}_r^{(M)}, \tag{4}$$

$$x_\tau = \left[w_{\tau,1}^{[1,1]}, w_{\tau,2}^{[1,1]}, \cdots, w_{\tau,d}^{[1,1]}\right]\begin{bmatrix} z_{II,1} \\ \vdots \\ z_{II,d} \end{bmatrix} + \left[w_{\tau,1}^{[1,2]}, w_{\tau,2}^{[1,2]}, \cdots, w_{\tau,d}^{[1,2]}\right]\begin{bmatrix} z_{I,1} \\ \vdots \\ z_{I,d} \end{bmatrix} + \left[ \phantom{aa} \cdots \phantom{a} z_{II,\lambda} \phantom{a} \cdots \phantom{aa} \right]\left[w_{\tau,\lambda,\mu}^{[2,1]}\right]\begin{bmatrix} \vdots \\ z_{II,\mu} \\ \vdots \end{bmatrix} +$$

$$\left[ \phantom{aa} \cdots \phantom{a} z_{I,\lambda} \phantom{a} \cdots \phantom{aa} \right]\left[w_{\tau,\lambda,\mu}^{[2,3]}\right]\begin{bmatrix} \vdots \\ z_{I,\mu} \\ \vdots \end{bmatrix} + \left[ \phantom{aa} \cdots \phantom{a} z_{I,\lambda} \phantom{a} \cdots \phantom{aa} \right]\left[w_{\tau,\lambda,\mu}^{[2,2]}\right]\begin{bmatrix} \vdots \\ z_{II,\mu} \\ \vdots \end{bmatrix}$$

Figure 5: Schematic for second order expansion with scalar output $x_\tau \in \mathbb{R}$. The abbreviations $z_{I,\lambda}, z_{I,\mu}$ are elements of $z_I$ with $\lambda, \mu \in [1, d]$. Similarly, $z_{II,\lambda}, z_{II,\mu}$ are elements of $z_{II}$. The first two terms (on the right side of the equation) are the first-order correlations; the next two terms are the second order auto-correlations. The last term expresses the second order cross-correlations.

where $\circ$ is the vector outer product. The factor matrices $\{ \boldsymbol{U}_{[m]} = [\boldsymbol{u}_1^{(m)}, \boldsymbol{u}_2^{(m)}, \cdots, \boldsymbol{u}_R^{(m)}] \in \mathbb{R}^{I_m \times R}\}_{m=1}^M$ collect the vectors from the rank-one components. By considering the mode-1 unfolding of $\mathcal{W}$, the CP decomposition can be written in matrix form as:

$$\boldsymbol{W}_{(1)} \doteq \boldsymbol{U}_{[1]} \left( \bigodot_{m=M}^{2} \boldsymbol{U}_{[m]} \right)^T \tag{5}$$

The following lemma is useful in our method:

**Lemma 1.** *For a set of $N$ matrices $\{\boldsymbol{A}_{[\nu]} \in \mathbb{R}^{I_\nu \times K}\}_{\nu=1}^N$ and $\{\boldsymbol{B}_{[\nu]} \in \mathbb{R}^{I_\nu \times L}\}_{\nu=1}^N$, the following equality holds:*

$$\left( \bigodot_{\nu=1}^{N} \boldsymbol{A}_{[\nu]} \right)^T \cdot \left( \bigodot_{\nu=1}^{N} \boldsymbol{B}_{[\nu]} \right) = (\boldsymbol{A}_{[1]}^T \cdot \boldsymbol{B}_{[1]}) * \ldots * (\boldsymbol{A}_{[N]}^T \cdot \boldsymbol{B}_{[N]}) \tag{6}$$

An indicative proof can be found in the Appendix of Chrysos et al. [2019].

## B.2 Scalar output

The proposed formulation expresses higher-order interactions of the input variables. To elaborate that, we develop the single output case below. That is, we focus on an element $\tau$ of the output vector, e.g., a single pixel. In the next few paragraphs, we consider the case of a scalar output $x_\tau$, with $\tau \in [1, o]$ when the input variables are $z_I, z_{II} \in \mathbb{K}^d$. To avoid cluttering the notation we only refer to the scalar output with $x_\tau$ in the next few paragraphs.

As a reminder, the polynomial of expansion order $N \in \mathbb{N}$ with output $\boldsymbol{x} \in \mathbb{R}^o$ has the form:

$$\boldsymbol{x} = G(\boldsymbol{z}_I, \boldsymbol{z}_{II}) = \sum_{n=1}^N \sum_{\rho=1}^{n+1} \left( \mathcal{W}^{[n,\rho]} \prod_{j=2}^{\rho} \times_j \boldsymbol{z}_I \prod_{\tau=\rho+1}^{n+1} \times_\tau \boldsymbol{z}_{II} \right) + \boldsymbol{\beta} \tag{7}$$

We assume a second order expansion ($N = 2$) and let $\tau$ denote an arbitrary scalar output of $\boldsymbol{x}$. The first order correlations can be expressed through the sums $\sum_{\lambda=1}^d w_{\tau,\lambda}^{[1,1]} z_{I,\lambda}$ and $\sum_{\lambda=1}^d w_{\tau,\lambda}^{[1,2]} z_{I,\lambda}$. The second order correlations include both auto- and cross-correlations. The tensors $\mathcal{W}^{[2,1]}$ and $\mathcal{W}^{[2,3]}$ capture the auto-correlations, while the tensor $\mathcal{W}^{[2,2]}$ captures the cross-correlations.

A pictorial representation of the correlations are captured in Fig. 5. Collecting all the terms in an equation, each output is expressed as:

$$x_\tau = \beta_\tau + \sum_{\lambda=1}^d \left[ w_{\tau,\lambda}^{[1,1]} z_{II,\lambda} + w_{\tau,\lambda}^{[1,2]} z_{I,\lambda} + \sum_{\mu=1}^d w_{\tau,\lambda,\mu}^{[2,1]} z_{II,\lambda} z_{II,\mu} + \sum_{\mu=1}^d w_{\tau,\lambda,\mu}^{[2,3]} z_{I,\lambda} z_{I,\mu} + \sum_{\mu=1}^d w_{\tau,\lambda,\mu}^{[2,2]} z_{I,\lambda} z_{II,\mu} \right] \tag{8}$$

where $\beta_\tau \in \mathbb{R}$. Notice that all the correlations of up to second order are captured in (8).

## B.3 Second order derivation for two-variable input

In all our derivations, the variables associated with the first input $z_I$ have an $I$ notation, e.g., $U_{[1,I]}$. Respectively for the second input $z_{II}$, the notation $II$ is used.

Even though (7) enables any order of expansion, the learnable parameters increase exponentially, therefore we can use a coupled factorization to reduce the parameters. Next, we derive the factorization for a second order expansion (i.e., $N = 2$) and then provide the recursive relationship that generalizes it for an arbitrary order.

**Second order derivation**: For a second order expansion (i.e., $N = 2$ in (1)), we factorize each parameter tensor $\mathcal{W}^{[n,\rho]}$. We assume a coupled CP decomposition for each parameter as follows:

- Let $W_{(1)}^{[1,1]} = CU_{[1,II]}^T$ and $W_{(1)}^{[1,2]} = CU_{[1,I]}^T$ be the parameters for $n = 1$.

- Let $W_{(1)}^{[2,1]} = C(U_{[2,II]} \odot U_{[1,II]})^T$ and $W_{(1)}^{[2,3]} = C(U_{[2,I]} \odot U_{[1,I]})^T$ capture the second order correlations of a single variable ($z_{II}$ and $z_I$ respectively).

- The cross-terms are expressed in $\mathcal{W}^{[2,2]} \times_2 z_I \times_3 z_{II}$. The output of the $\tau$ element[4] is $\sum_{\lambda,\mu=1}^d w_{\tau,\lambda,\mu}^{[2,2]} z_{I,\lambda} z_{II,\mu}$. The product $\hat{\mathcal{W}}^{[2,2]} \times_2 z_{II} \times_3 z_I$ also results in the same elementwise expression. Hence, to allow for symmetric expression, we factorize the term $W_{(1)}^{[2,2]}$ as the sum of the two terms $C(U_{[2,II]} \odot U_{[1,I]})^T$ and $C(U_{[2,I]} \odot U_{[1,II]})^T$. For each of the two terms, we assume that the vector-valued inputs are accordingly multiplied.

The parameters $C \in \mathbb{R}^{o \times k}, U_{[m,\phi]} \in \mathbb{R}^{d \times k}$ ($m = 1, 2$ and $\phi = \{I, II\}$) are learnable. The aforementioned factorization results in the following equation:

$$x = CU_{[1,II]}^T z_{II} + CU_{[1,I]}^T z_I + C\left(U_{[2,II]} \odot U_{[1,II]}\right)^T \left(z_{II} \odot z_{II}\right) + C\left(U_{[2,I]} \odot U_{[1,I]}\right)^T \left(z_I \odot z_I\right) +$$
$$C\left(U_{[2,I]} \odot U_{[1,II]}\right)^T \left(z_I \odot z_{II}\right) + C\left(U_{[2,II]} \odot U_{[1,I]}\right)^T \left(z_{II} \odot z_I\right) + \beta \tag{9}$$

This expansion captures the correlations (up to second order) of the two input variables $z_I, z_{II}$.

To make the proof more complete, we remind the reader that the recursive relationship (i.e., (2) in the main paper) is:

$$x_n = x_{n-1} + \left(U_{[n,I]}^T z_I + U_{[n,II]}^T z_{II}\right) * x_{n-1} \tag{10}$$

for $n = 2, \ldots, N$ with $x_1 = U_{[1,I]}^T z_I + U_{[1,II]}^T z_{II}$ and $x = Cx_N + \beta$.

**Claim 1.** *The equation (9) is a special format of a polynomial that is visualized as in Fig. 1 of the main paper. Equivalently, prove that (9) follows the recursive relationship of (10).*

*Proof.* We observe that the first two terms of (9) are equal to $Cx_1$ (from (10)). By applying Lemma 1 in the terms that have Khatri-Rao product, we obtain:

$$x = \beta + Cx_1 + C\left\{\left(U_{[2,II]}^T z_{II}\right) * \left(U_{[1,II]}^T z_{II}\right) + \left(U_{[2,I]}^T z_I\right) * \left(U_{[1,I]}^T z_I\right) +\right.$$
$$\left(U_{[2,I]}^T z_I\right) * \left(U_{[1,II]}^T z_{II}\right) + \left(U_{[2,II]}^T z_{II}\right) * \left(U_{[1,I]}^T z_I\right)\right\} = \tag{11}$$
$$\beta + Cx_1 + C\left\{\left[\left(U_{[2,I]}^T z_I\right) + \left(U_{[2,II]}^T z_{II}\right)\right] * x_1\right\} = Cx_2 + \beta$$

The last equation is precisely the one that arises from the recursive relationship from (10).

$\square$

---

[4] An elementwise analysis (with a scalar output) is provided on the supplementary (sec. B.2).

To prove the recursive formula for the $N^{th}$ order expansion, a similar pattern as in sec.C of Poly-GAN [Chrysos et al., 2019] can be followed. Specifically, the difference here is that because of the two input variables, the auto- and cross-correlation variables should be included. Other than that, the same factor sharing is followed.

## B.4 Nested-CoPE model for two-variable input

The model proposed above (i.e., (10)), relies on a single coupled CP decomposition, however a more flexible model can factorize each level with a CP decomposition. To effectively do that, we utilize learnable hyper-parameters $\boldsymbol{b}_{[n]} \in \mathbb{R}^\omega$ for $n \in [1, N]$, which act as scaling factors for each parameter tensor. Then, a polynomial of expansion order $N \in \mathbb{N}$ with output $\boldsymbol{x} \in \mathbb{R}^o$ has the form:

$$\boldsymbol{x} = G(\boldsymbol{z}_{\mathrm{I}}, \boldsymbol{z}_{\mathrm{II}}) = \sum_{n=1}^{N} \sum_{\rho=2}^{n+2} \left( \boldsymbol{\mathcal{W}}^{[n,\rho-1]} \times_2 \boldsymbol{b}_{[N+1-n]} \prod_{j=3}^{\rho} \times_j \boldsymbol{z}_{\mathrm{I}} \prod_{\tau=\rho+1}^{n+2} \times_\tau \boldsymbol{z}_{\mathrm{II}} \right) + \boldsymbol{\beta} \qquad (12)$$

To demonstrate the factorization without cluttering the notation, we assume a second order expansion in (12).

**Second order derivation**: The second order expansion, i.e., $N = 2$, is derived below. We jointy factorize all parameters of (12) with a nested decomposition as follows:

- First order parameters : $\boldsymbol{W}_{(1)}^{[1,1]} = \boldsymbol{C}(\boldsymbol{A}_{[2,II]} \odot \boldsymbol{B}_{[2]})^T$ and $\boldsymbol{W}_{(1)}^{[1,2]} = \boldsymbol{C}(\boldsymbol{A}_{[2,I]} \odot \boldsymbol{B}_{[2]})^T$.

- Let $\boldsymbol{W}_{(1)}^{[2,1]} = \boldsymbol{C}\left\{ \boldsymbol{A}_{[2,II]} \odot \left[ \left( \boldsymbol{A}_{[1,II]} \odot \boldsymbol{B}_{[1]} \right) \boldsymbol{V}_{[2]} \right] \right\}^T$ and $\boldsymbol{W}_{(1)}^{[2,3]} = \boldsymbol{C}\left\{ \boldsymbol{A}_{[2,I]} \odot \left[ \left( \boldsymbol{A}_{[1,I]} \odot \boldsymbol{B}_{[1]} \right) \boldsymbol{V}_{[2]} \right] \right\}^T$ capture the second order correlations of a single variable ($\boldsymbol{z}_{\mathrm{II}}$ and $\boldsymbol{z}_{\mathrm{I}}$ respectively).

- The cross-terms are included in $\boldsymbol{\mathcal{W}}^{[2,2]} \times_2 \boldsymbol{b}_{[1]} \times_3 \boldsymbol{z}_{\mathrm{I}} \times_4 \boldsymbol{z}_{\mathrm{II}}$. The output of the $\tau$ element is expressed as $\sum_{\nu=1}^{\omega} \sum_{\lambda,\mu=1}^{d} w_{\tau,\nu,\lambda,\mu}^{[2,2]} b_{[1],\omega} z_{\mathrm{I},\lambda} z_{\mathrm{II},\mu}$. Similarly, the product $\hat{\boldsymbol{\mathcal{W}}}^{[2,2]} \times_2 \boldsymbol{b}_{[1]} \times_3 \boldsymbol{z}_{\mathrm{II}} \times_4 \boldsymbol{z}_{\mathrm{I}}$ has output $\sum_{\nu=1}^{\omega} \sum_{\lambda,\mu=1}^{d} w_{\tau,\nu,\mu,\lambda}^{[2,2]} b_{[1],\omega} z_{\mathrm{I},\lambda} z_{\mathrm{II},\mu}$ for the $\tau$ element. Notice that the only change in the two expressions is the permutation of the third and forth modes of the tensor; the rest of the expression remains the same. Therefore, to account for this symmetry we factorize the term $\boldsymbol{\mathcal{W}}^{[2,2]}$ as the sum of two terms and assume that each term is multiplied by the respective terms. Let $\boldsymbol{W}_{(1)}^{[2,2]} = \boldsymbol{C}\left\{ \boldsymbol{A}_{[2,I]} \odot \left[ \left( \boldsymbol{A}_{[1,II]} \odot \boldsymbol{B}_{[1]} \right) \boldsymbol{V}_{[2]} \right] + \boldsymbol{A}_{[2,II]} \odot \left[ \left( \boldsymbol{A}_{[1,I]} \odot \boldsymbol{B}_{[1]} \right) \boldsymbol{V}_{[2]} \right] \right\}^T$.

The parameters $C \in \mathbb{R}^{o \times k}$, $A_{[n,\phi]} \in \mathbb{R}^{d \times k}$, $V_{[n]} \in \mathbb{R}^{k \times k}$, $B_{[n]} \in \mathbb{R}^{\omega \times k}$ for $n = 1, 2$ and $\phi = \{I, II\}$ are learnable. Collecting all the terms above and extracting $C$ as a common factor (we ommit $C$ below to avoid cluttering the notation):

$$
\begin{aligned}
& (A_{[2,II]} \odot B_{[2]})^T (z_{\mathrm{II}} \odot b_{[2]}) + (A_{[2,I]} \odot B_{[2]})^T (z_{\mathrm{I}} \odot b_{[2]}) + \\
& \left\{ A_{[2,II]} \odot \left[ \left( A_{[1,II]} \odot B_{[1]} \right) V_{[2]} \right] \right\}^T (z_{\mathrm{II}} \odot z_{\mathrm{II}} \odot b_{[1]}) + \\
& \left\{ A_{[2,I]} \odot \left[ \left( A_{[1,I]} \odot B_{[1]} \right) V_{[2]} \right] \right\}^T (z_{\mathrm{I}} \odot z_{\mathrm{I}} \odot b_{[1]}) + \\
& \left\{ A_{[2,I]} \odot \left[ \left( A_{[1,II]} \odot B_{[1]} \right) V_{[2]} \right] \right\}^T (z_{\mathrm{I}} \odot z_{\mathrm{II}} \odot b_{[1]}) + \\
& \left\{ A_{[2,II]} \odot \left[ \left( A_{[1,I]} \odot B_{[1]} \right) V_{[2]} \right] \right\}^T (z_{\mathrm{II}} \odot z_{\mathrm{I}} \odot b_{[1]}) = \\
& \qquad \left( A_{[2,II]}^T z_{\mathrm{II}} + A_{[2,I]}^T z_{\mathrm{I}} \right) * \left( B_{[2]}^T b_{[2]} \right) + \\
& \left( A_{[2,II]}^T z_{\mathrm{II}} + A_{[2,I]}^T z_{\mathrm{I}} \right) * \left\{ V_{[2]}^T \left[ \left( A_{[1,II]}^T z_{\mathrm{II}} + A_{[1,I]}^T z_{\mathrm{I}} \right) * \left( B_{[1]}^T b_{[1]} \right) \right] \right\}
\end{aligned}
\tag{13}
$$

The last equation is precisely a recursive equation that can be expressed with the Fig. 6 or equivalently the generalized recursive relationship below.

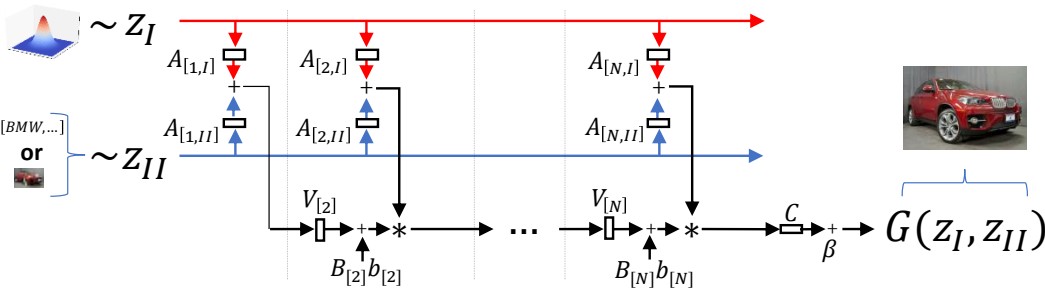

Figure 6: Abstract schematic for $N^{th}$ order approximation of $x = G(z_I, z_{II})$ with Nested-CoPE model. The inputs $z_I, z_{II}$ are symmetric in our formulation. We denote with $z_I$ a sample from the noise distribution (e.g., Gaussian), while $z_{II}$ symbolizes a sample from a conditional input (e.g., a class label or a low-resolution image).

**Recursive relationship**: The recursive formula for the Nested-CoPE model with arbitrary expansion order $N \in \mathbb{N}$ is the following:

$$
x_n = \left( A_{[n,I]}^T z_{\mathrm{I}} + A_{[n,II]}^T z_{\mathrm{II}} \right) * \left( V_{[n]}^T x_{n-1} + B_{[n]}^T b_{[n]} \right)
\tag{14}
$$

where $n \in [2, N]$ and $x_1 = \left( A_{[1,I]}^T z_{\mathrm{I}} + A_{[1,II]}^T z_{\mathrm{II}} \right) * \left( B_{[1]}^T b_{[1]} \right)$. The parameters $C \in \mathbb{R}^{o \times k}$, $A_{[n,\phi]} \in \mathbb{R}^{d \times k}$, $V_{[n]} \in \mathbb{R}^{k \times k}$, $B_{[n]} \in \mathbb{R}^{\omega \times k}$ for $\phi = \{I, II\}$ are learnable. Then, the output $x = C x_N + \beta$.

The Nested-CoPE model manifests an alternative network that relies on slightly modified assumptions on the decomposition. Thus, changing the underlying assumptions of the decomposition can modify the resulting network. This can be an important tool for domain-specific applications, e.g., when the domain-knowledge should be inserted in the last layers.

## C    Beyond two variables

Frequently, more than one conditional inputs are required [Yu et al., 2018b, Xu et al., 2017, Maximov et al., 2020]. In such tasks, the aforementioned framework can be generalized to more than two input variables. We demonstrate how this is possible with three variables; then it can trivially extended to an arbitrary number of input variables.

Let $z_\mathrm{I}, z_\mathrm{II}, z_\mathrm{III} \in \mathbb{K}^d$ denote the three input variables. We aim to learn a function that captures the higher-order interactions of the input variables. The polynomial of expansion order $N \in \mathbb{N}$ with output $\boldsymbol{x} \in \mathbb{R}^o$ has the form:

$$\boldsymbol{x} = G(\boldsymbol{z}_\mathrm{I}, \boldsymbol{z}_\mathrm{II}, \boldsymbol{z}_\mathrm{III}) = \sum_{n=1}^{N} \sum_{\rho=1}^{n+1} \sum_{\delta=\rho}^{n+1} \left( \boldsymbol{\mathcal{W}}^{[n,\rho,\delta]} \prod_{j=2}^{\rho} \times_j \boldsymbol{z}_\mathrm{I} \prod_{\tau=\rho+1}^{\delta} \times_\tau \boldsymbol{z}_\mathrm{II} \prod_{\zeta=\delta+1}^{n+1} \times_\zeta \boldsymbol{z}_\mathrm{III} \right) + \boldsymbol{\beta} \quad (15)$$

where $\boldsymbol{\beta} \in \mathbb{R}^o$ and $\boldsymbol{\mathcal{W}}^{[n,\rho,\delta]} \in \mathbb{R}^{o \times \prod_{m=1}^{n} \times_m d}$ (for $n \in [1, N]$ and $\rho, \delta \in [1, n+1]$) are the learnable parameters. As in the two-variable input, the unknown parameters increase exponentially. To that end, we utilize a joint factorization with factor sharing. The recursive relationship of such a factorization is:

$$\boldsymbol{x}_n = \boldsymbol{x}_{n-1} + \left( \boldsymbol{U}_{[n,I]}^T \boldsymbol{z}_\mathrm{I} + \boldsymbol{U}_{[n,II]}^T \boldsymbol{z}_\mathrm{II} + \boldsymbol{U}_{[n,III]}^T \boldsymbol{z}_\mathrm{III} \right) * \boldsymbol{x}_{n-1} \quad (16)$$

for $n = 2, \ldots, N$ with $\boldsymbol{x}_1 = \boldsymbol{U}_{[1,I]}^T \boldsymbol{z}_\mathrm{I} + \boldsymbol{U}_{[1,II]}^T \boldsymbol{z}_\mathrm{II} + \boldsymbol{U}_{[1,III]}^T \boldsymbol{z}_\mathrm{III}$ and $\boldsymbol{x} = \boldsymbol{C}\boldsymbol{x}_N + \boldsymbol{\beta}$.

Notice that the pattern (for each order) is similar to the two-variable input: a) a different embedding is found for each input variable, b) the embeddings are added together, c) the result is multiplied elementwise with the representation of the previous order.

## D    Concatenation of inputs

A popular method used for conditional generation is to concatenate the conditional input with the noise labels. However, as we showcase below, concatenation has two significant drawbacks when compared to our framework. To explain those, we will define a concatenation model.

Let $z_\mathrm{I} \in \mathbb{K}_1^{d_1}, z_\mathrm{II} \in \mathbb{K}_2^{d_2}$ where $\mathbb{K}_1, \mathbb{K}_2$ can be a subset of real or natural numbers. The output of a concatenation layer is $x = \boldsymbol{P}^T \left[ z_\mathrm{I}; z_\mathrm{II} \right]^T$ where the symbol ';' denotes the concatenation and $\boldsymbol{P} \in \mathbb{R}^{(d_1+d_2) \times o}$ is an affine transformation on the concatenated vector. The $j^{th}$ output is $x_j = \sum_{\tau=1}^{d_1} p_{\tau,j} z_{\mathrm{I},\tau} + \sum_{\tau=1}^{d_2} p_{\tau+d_1,j} z_{\mathrm{II},\tau}$.

Therefore, the two differences from the concatenation case are:

- If the input variables are concatenated together we obtain an additive format, not a multiplicative that can capture cross-term correlations. That is, the multiplicative format does allow achieving higher-order auto- and cross- term correlations.

- The concatenation changes the dimensionality of the embedding space. Specifically, the input space has dimensionality $d_1 + d_2$. That has a significant toll on the size of the filters (i.e., it increases the learnable parameters), while still having an additive impact. On the contrary, our framework does not change the dimensionality of the embedding spaces.

## E    In-depth differences from $\Pi$-Net

In the next few paragraphs, we conduct an in-depth analysis of the differences between $\Pi$-Net and CoPE. The analysis assumes knowledge of the proposed model, i.e., (2).

Chrysos et al. [2020] introduce $\Pi$-Net as a polynomial expansion of a single input variable. Their goal is to model functions $\boldsymbol{x} = G(\boldsymbol{z})$ as high-order polynomial expansions of $\boldsymbol{z}$. Their focus is towards using a single-input variable $\boldsymbol{z}$, which can be noise in case of image generation or an image

in discriminative experiments. The authors express the StyleGAN architecture [Karras et al., 2019] as a polynomial expansion, while they advocate that the impressive results can be attributed to the polynomial expansion.

To facilitate the in-depth analysis, the recursive relationship that corresponds to (2) is provided below. An $N^{th}$ order expansion in $\Pi$-Net is expressed as:

$$\boldsymbol{x}_n = \left(\boldsymbol{\Lambda}_{[n]}^T \boldsymbol{z}\right) * \boldsymbol{x}_{n-1} + \boldsymbol{x}_{n-1} \tag{17}$$

for $n = 2, \ldots, N$ with $\boldsymbol{x}_1 = \boldsymbol{\Lambda}_{[1]}^T \boldsymbol{z}$ and $\boldsymbol{x} = \boldsymbol{\Gamma} \boldsymbol{x}_N + \boldsymbol{\beta}$. The parameters $\boldsymbol{\Lambda}, \boldsymbol{\Gamma}$ are learnable.

In this work, we focus on conditional data generation, i.e., there are multiple input variables available as auxiliary information. The trivial application of $\Pi$-Net would be to concatenate all the $M$ input variables $\boldsymbol{z}_{\text{I}}, \boldsymbol{z}_{\text{II}}, \boldsymbol{z}_{\text{III}}, \ldots$. The input variable $\boldsymbol{z}$ becomes $\boldsymbol{z} = \left[\boldsymbol{z}_{\text{I}}; \boldsymbol{z}_{\text{II}}; \boldsymbol{z}_{\text{III}}; \ldots\right]$, where the symbol ';' denotes the concatenation. Then, the polynomial expansion of $\Pi$-Net can be learned on the concatenated $\boldsymbol{z}$. However, there are four significant reasons that we believe that this is not as flexible as the proposed CoPE.

When we refer to $\Pi$-Net below, we refer to the model with concatenated input. In addition, let $\boldsymbol{z}_{\text{I}} \in \mathbb{K}_1^{d_1}, \boldsymbol{z}_{\text{II}} \in \mathbb{K}_2^{d_2}$ denote the input variables where $\mathbb{K}_1, \mathbb{K}_2$ can be a subset of real or natural numbers.

**Parameter sharing:** CoPE allows additional flexibility in the structure of the architecture, since CoPE utilizes a different projection layer for each input variable. We utilize this flexibility to share the parameters of the conditional input variable; as we detail in (19), we set $\boldsymbol{U}_{[n,II]} = \boldsymbol{U}_{[1,II]}$ on (2). If we want to perform a similar sharing in $\Pi$-Net, the formulation equivalent to (17) would be $(\lambda_{[n]})_i = (\lambda_{[1]})_i$ for $i = d_1, \ldots, d_1 + d_2$. However, sharing only part of the matrix might be challenging. Additionally, when $\boldsymbol{\Lambda}$ is a convolution, the sharing pattern is not straightforward to be computed. Therefore, CoPE enables additional flexibility to the model, which is hard to be included in $\Pi$-Net.

**Inductive bias:** The inductive bias is crucial in machine learning [Zhao et al., 2018], however concatenating the variables restricts the flexibility of the model (i.e. $\Pi$-Net). To illustrate that, let us use the super-resolution experiments as an example. The input variable $\boldsymbol{z}_{\text{I}}$ is the noise vector and $\boldsymbol{z}_{\text{II}}$ is the (vectorized) low-resolution image. If we concatenate the two variables, then we should use a fully-connected (dense) layer, which does not model well the spatial correlations. Instead, with CoPE, we use a fully-connected layer for the noise vector and a convolution for $\boldsymbol{z}_{\text{II}}$ (low-resolution image). The convolution reduces the number of parameters and captures the spatial correlations in the image. Thus, by concatenating the variables, we reduce the flexibility of the model.

**Dimensionality of the inputs:** The dimensionality of the inputs might vary orders of magnitude, which might create an imbalance during learning. For instance, in class-conditional generation concatenating the one-hot labels in the input does not scale well when there are hundreds of classes [Odena et al., 2017]. We observe a similar phenomenon in class-conditional generation: in Cars196 (with 196 classes) the performance of $\Pi$-Net deteriorates considerably when compared to its (relative) performance in CIFAR10 (with 10 classes). On the contrary, CoPE does not fuse the elements of the input variables directly, but it projects them into a subspace appropriate for adding them.

**Order of expansion with respect to each variable:** Frequently, the two inputs do not require the same order of expansion. Without loss of generality, assume that we need correlations up to $N_I$ and $N_{II}$ order (with $N_I < N_{II}$) from $\boldsymbol{z}_{\text{I}}$ and $\boldsymbol{z}_{\text{II}}$ respectively. CoPE includes a different transformation for each variable, i.e., $\boldsymbol{U}_{[n,I]}$ for $\boldsymbol{z}_{\text{I}}$ and $\boldsymbol{U}_{[n,II]}$ for $\boldsymbol{z}_{\text{II}}$. Then, we can set $\boldsymbol{U}_{[n,I]} = 0$ for $n > N_I$. On the contrary, the concatenation of inputs (in $\Pi$-Net) constrains the expansion to have the same order with respect to each variable.

All in all, we can use concatenation to fuse variables and use $\Pi$-Net, however an inherently multivariate model is more flexible and can better encode the types of inductive bias required for conditional data generation.

# F Differences from other networks cast as polynomial neural networks

A number of networks with impressive results have emerged in (conditional) data generation the last few years. Three such networks that are particularly interesting in our context are Karras et al. [2019], Park et al. [2019], Chen et al. [2019]. We analyze below each method and how it relates to polynomial expansions:

- Karras et al. [2019] propose an Adaptive instance normalization (AdaIN) method for unsupervised image generation. An AdaIN layer expresses a second-order interaction[5]: $\boldsymbol{h} = (\boldsymbol{\Lambda}^T \boldsymbol{w}) * n(c(\boldsymbol{h}_{in}))$, where $n$ is a normalization, $c$ the convolution operator and $\boldsymbol{w}$ is the transformed noise $\boldsymbol{w} = MLP(\boldsymbol{z}_\mathrm{I})$ (mapping network). The parameters $\boldsymbol{\Lambda}$ are learnable, while $\boldsymbol{h}_{in}$ is the input to the AdaIN. Stacking AdaIN layers results in a polynomial expansion with a single variable.

- Chen et al. [2019] propose a normalization method, called sBN, to stabilize the GAN training. The method performs a 'self-modulation' with respect to the noise variable and optionally the conditional variable in the class-conditional generation setting. Henceforth, we focus on the class-conditional setting that is closer to our work. sBN injects the network layers with a multiplicative interaction of the input variables. Specifically, sBN projects the conditional variable into the space of the variable $\boldsymbol{z}_\mathrm{I}$ through an embedding function. Then, the interaction of the two vector-like variables is passed through a fully-connected layer (and a ReLU activation function); the result is injected into the network through the batch normalization parameters. If we cast sBN as a polynomial expansion, it expresses a single polynomial expansion with respect to the input noise and the input conditional variable[6].

- Park et al. [2019] introduce a spatially-adaptive normalization, i.e., SPADE, to improve semantic image synthesis. Their model, referred to as SPADE in the remainder of this work, assumes a semantic layout as a conditional input that facilitates the image generation. We analyze in sec. F.1 how to obtain the formulation of their spatially-adaptive normalization. If we cast SPADE as a polynomial expansion, it expresses a polynomial expansion with respect to the conditional variable.

The aforementioned works propose or modify the batch normalization layer to improve the performance or stabilize the training, while in our work we propose the multivariate polynomial as a general function approximation technique for conditional data generation. Nevertheless, given the interpretation of the previous works in the perspective of polynomials, we still can express them as special cases of MVP. Methodologically, there are **two significant limitations** that none of the aforementioned works tackle:

- The aforementioned architectures focus on no or one conditional variable. On the contrary, CoPE naturally extends to **arbitrarily many conditional variables**.

- Even though the aforementioned three architectures use (implicitly) a polynomial expansion, a significant factor is the order of the expansion. In our work, the **product of polynomials** enables capturing higher-order correlations without increasing the amount of layers substantially (sec. 3.2).

In addition to the aforementioned methodological differences, *our work is the only polynomial expansion that conducts experiments on a variety of conditional data generation tasks*. Thus, we both demonstrate methodologically and verify experimentally that CoPE can be used for a wide range of conditional data generation tasks.

## F.1 In-depth differences from SPADE

In the next few paragraphs, we conduct an in-depth analysis of the differences between SPADE and CoPE.

---

[5]The formulation is derived from the public implementation of the authors.

[6]In CoPE, we do not learn a single embedding function for the conditional variable. In addition, we do not project the (transformed) conditional variable to the space of the noise-variable. Both of these can be achieved by making simplifying assumptions on the factor matrices of CoPE.

Park et al. [2019] introduce a spatially-adaptive normalization, i.e., SPADE, to improve semantic image synthesis. Their model, referred to as SPADE in the remainder of this work, assumes a semantic layout as a conditional input that facilitates the image generation.

The $n^{th}$ model block applies a normalization on the representation $\boldsymbol{x}_{n-1}$ of the previous layer and then it performs an elementwise multiplication with a transformed semantic layout. The transformed semantic layout can be denoted as $\boldsymbol{A}_{[n,II]}^T \boldsymbol{z}_{II}$ where $\boldsymbol{z}_{II}$ denotes the conditional input to the generator. The output of this elementwise multiplication is then propagated to the next model block that performs the same operations. Stacking $N$ such blocks results in an $N^{th}$ order polynomial expansion which is expressed as:

$$\boldsymbol{x}_n = \left( \boldsymbol{A}_{[n,II]}^T \boldsymbol{z}_{II} \right) * \left( \boldsymbol{V}_{[n]}^T \boldsymbol{x}_{n-1} + \boldsymbol{B}_{[n]}^T \boldsymbol{b}_{[n]} \right) \tag{18}$$

where $n \in [2, N]$ and $\boldsymbol{x}_1 = \boldsymbol{A}_{[1,I]}^T \boldsymbol{z}_I$. The parameters $\boldsymbol{C} \in \mathbb{R}^{o \times k}, \boldsymbol{A}_{[n,\phi]} \in \mathbb{R}^{d \times k}, \boldsymbol{V}_{[n]} \in \mathbb{R}^{k \times k}, \boldsymbol{B}_{[n]} \in \mathbb{R}^{\omega \times k}$ for $\phi = \{I, II\}$ are learnable. Then, the output $\boldsymbol{x} = \boldsymbol{C}\boldsymbol{x}_N + \boldsymbol{\beta}$.

SPADE as expressed in (18) resembles one of the proposed models of CoPE (specifically (14)). In particular, it expresses a polynomial with respect to the conditional variable. The parameters $\boldsymbol{A}_{[n,I]}$ are set as zero, which means that there are no higher-order correlations with respect to the input variable $\boldsymbol{z}_I$. Therefore, our work bears the following differences from Park et al. [2019]:

- SPADE proposes a normalization scheme that is only applied to semantic image generation. On the contrary, our proposed CoPE can be applied to any conditional data generation task, e.g., class-conditional generation or image-to-image translation.

- **SPADE is a special case of CoPE**. In particular, by setting i) $\boldsymbol{A}_{[1,II]}$ equal to zero, ii) $\boldsymbol{A}_{[n,I]}$ in (14) equal to zero, we obtain SPADE. In addition, CoPE allows different assumptions on the decompositions which lead to an alternative structure, such as (2).

- SPADE proposes a polynomial expansion with respect to a single variable. On the other hand, our model can extend to an arbitrary number of input variables to account for auxiliary labels, e.g., (16).

- Even though SPADE models higher-order correlations of the conditional variable, it still does not leverage the higher-order correlations of the representations (e.g., as in the product of polynomials) and hence without activation functions it might not work as well as the two-variable expansion.

Park et al. [2019] exhibit impressive generation results with large-scale computing (i.e., they report results using NVIDIA DGX with 8 V100 GPUs). Our goal is not to compete in computationally heavy, large-scale experiments, but rather to illustrate the benefits of the generic formulation of CoPE.

SPADE is an important baseline for our work. In particular, we augment SPADE in wo ways: a) by extending it to accept both continuous and discrete variables in $\boldsymbol{z}_{II}$ and b) by adding polynomial terms with respect to the input variable $\boldsymbol{z}_I$. The latter model is referred to as SPADE-CoPE (details on the next section).

## G  Experimental details

**Metrics:**  The two most popular metrics [Lucic et al., 2018, Creswell et al., 2018] for evaluation of the synthesized images are the Inception Score (IS) [Salimans et al., 2016] and the Frechet Inception Distance (FID) [Heusel et al., 2017]. The metrics utilize the pretrained Inception network [Szegedy et al., 2015] to extract representations of the synthesized images. FID assumes that the representations extracted follow a Gaussian distribution and matches the statistics (i.e., mean and variance) of the representations between real and synthesized samples. Alternative evaluation metrics have been reported as inaccurate, e.g., in Theis et al. [2016], thus we use the IS and FID. Following the standard practice of the literature, the IS is computed by synthesizing $5,000$ samples, while the FID is computed using $10,000$ samples.

The IS is used exclusively for images of natural scenes as a metric. The reasoning behind that is that the Inception network has been trained on images of natural scenes. On the contrary, the FID metric

relies on the first and second-order moments of the representations, which are considered more robust to different types of images. Hence, we only report IS for the CIFAR10 related experiments, while for the rest the FID is reported.

**Dataset details:**    There are eight datasets used in this work:

- Large-scale CelebFaces Attributes (or *CelebA* for short) [Liu et al., 2015] is a large-scale face attributes dataset with $202,000$ celebrity images. We use $160,000$ images for training our method.

- *Cars196* [Krause et al., 2013] is a dataset that includes different models of cars in different positions and backgrounds. Cars196 has $16,000$ images, while the images have substantially more variation than CelebA faces.

- *CIFAR10* [Krizhevsky et al., 2014] contains $60,000$ images of natural scenes. Each image is of resolution $32 \times 32 \times 3$ and is classified in one of the $10$ classes. CIFAR10 is frequently used as a benchmark for image generation.

- The Street View House Numbers dataset (or *SVHN* for short) [Netzer et al., 2011] has $100,000$ images of digits ($73,257$ of which for training). SVHN includes color house-number images which are classified in 10 classes; each class corresponds to a digit 0 to 9. SVHN images are diverse (e.g., with respect to background, scale).

- *MNIST* [LeCun et al., 1998] consists of images with handwritten digits. Each images depicts a single digit (annotated from 0 to 9) in a $28 \times 28$ resolution. The dataset includes $60,000$ images for training.

- *Shoes* [Yu and Grauman, 2014, Xie and Tu, 2015] consists of $50,000$ images of shoes, where the edges of each shoe are extracted [Isola et al., 2017].

- *Handbags* [Zhu et al., 2016, Xie and Tu, 2015] consists of more than $130,000$ images of handbag items. The edges have been computed for each image and used as conditional input to the generator [Isola et al., 2017].

- *Anime characters* dataset [Jin et al., 2017] consists of anime characters that are generated based on specific attributes, e.g., hair color. The public version used[7] contains annotations on the hair color and the eye color. We consider 7 classes on the hair color and 6 classes on the eye color, with a total of $14,000$ training images.

All the images of CelebA, Cars196, Shoes and Handbags are resized to $64 \times 64$ resolution.

**Architectures:**    The discriminator structure is left the same for each experiment, we focus only on the generator architecture. All the architectures are based on two different generator schemes, i.e., the SNGAN [Miyato and Koyama, 2018] and the polynomial expansion of Chrysos et al. [2020] that does not include activation functions in the generator.

The variants of the generator of SNGAN are described below:

- **SNGAN** [Miyato and Koyama, 2018]: The generator consists of a convolution, followed by three residual blocks. The discriminator is also based on successive residual blocks. The public implementation of SNGAN with conditional batch normalization (CBN) is used as the baseline.

- **SNGAN-CoPE** [proposed]: We convert the resnet-based generator of SNGAN to an CoPE model. To obtain CoPE, the SNGAN is modified in two ways: a) the Conditional Batch Normalization (CBN) is converted into batch normalization [Ioffe and Szegedy, 2015], b) the injections of the two embeddings (from the inputs) are added after each residual block, i.e. the formula of (2). In other words, the generator is converted to a product of two-variable polynomials.

- **SNGAN-CONC**: Based on SNGAN-CoPE, we replace each Hadamard product with a concatenation. This implements the variant mentioned in sec. D.

---

[7]The version is downloaded following the instructions of `https://github.com/bchao1/Anime-Generation`.

- **SNGAN-SPADE** [Park et al., 2019]: As described in sec. F.1, SPADE is a polynomial with respect to the conditional variable $z_{II}$. The generator of SNGAN-CoPE is modified to perform the Hadamard product with respect to the conditional variable every time.

The variants of the generator of Π-Net are described below:

- **Π-Net** [Chrysos et al., 2020]: The generator is based on a product of polynomials. The first polynomials use fully-connected connections, while the next few polynomials use cross-correlations. The discriminator is based on the residual blocks of SNGAN. We stress out that the generator does not include any activation functions apart from a hyperbolic tangent in the output space for normalization. The authors advocate that this exhibits the expressivity of the designed model.

- **Π-Net-SICONC**: The generator structure is based on Π-Net with two modifications: a) the Conditional Batch Normalization is converted into batch normalization [Ioffe and Szegedy, 2015], b) the second-input is concatenated with the first (i.e., the noise) in the input of the generator. Thus, this is a single variable polynomial, i.e., a Π-Net, where the second-input is vectorized and concatenated with the first. This baseline implements the Π-Net described in sec. E.

- **CoPE** [proposed]: The generator of Π-Net is converted to an CoPE model with two modifications: a) the Conditional Batch Normalization is converted into batch normalization [Ioffe and Szegedy, 2015], b) instead of having a Hadamard product with a single variable as in Π-Net, the formula with the two-variable input (e.g., (2)) is followed.

- **GAN-CONC**: Based on CoPE, each Hadamard product is replaced by a concatenation. This implements the variant mentioned in sec. D.

- **GAN-ADD**: Based on CoPE, each Hadamard product is replaced by an addition. This modifies (14) to $x_n = \left( A_{[n,I]}^T z_I + A_{[n,II]}^T z_{II} \right) + \left( V_{[n]}^T x_{n-1} + B_{[n]}^T b_{[n]} \right)$.

- **SPADE** [Park et al., 2019]: As described in sec. F.1, SPADE defines a polynomial with respect to the conditional variable $z_{II}$. The generator of Π-Net is modified to perform the Hadamard product with respect to the conditional variable every time.

- **SPADE-CoPE** [proposed]: This is a variant we develop to bridge the gap between SPADE and the proposed CoPE. Specifically, we augment the aforementioned SPADE twofold: a) the dense layers in the input space are converted into a polynomial with respect to the variable $z_I$ and b) we also convert the polynomial in the output (i.e., the rightmost polynomial in the Fig. 7 schematics) to a polynomial with respect to the variable $z_I$. This model captures higher-order correlations of the variable $z_I$ that SPADE did not not originally include. This model still includes single variable polynomials, however the input in each polynomial varies and is not only the conditional variable.

We should note that StyleGAN is a special case of Π-Net as demonstrated by Chrysos et al. [2020], and converting it into a products of polynomials formulation improves its performance. Hence, we use directly the Π-Net. The two baselines GAN-CONC and GAN-ADD capture only additive correlations, hence they cannot effectively model complex distributions without activation functions. Nevertheless, they are added as a reference point to emphasize the benefits of higher-order polynomial expansions.

An abstract schematic of the generators that are in the form of products of polynomials is depicted in Fig. 7. Notice that the compared methods from the literature use polynomials of a single variable, while we propose a polynomial with an arbitrary number of inputs (e.g., two-input shown in the schematic).

**Implementation details of CoPE:**   Throughout this work, we reserve the symbol $z_{II}$ for the conditional input (e.g., a class label). In each polynomial, we reduce further the parameters by using the same embedding for the conditional variables. That is expressed as:

$$U_{[n,II]} = U_{[1,II]} \tag{19}$$

for $n = 2, \ldots, N$. Equivalently, that would be $A_{[n,II]} = A_{[1,II]}$ in (14). Additionally, Nested-CoPE performed better in our preliminary experiments, thus we use (14) to design each polynomial. Given

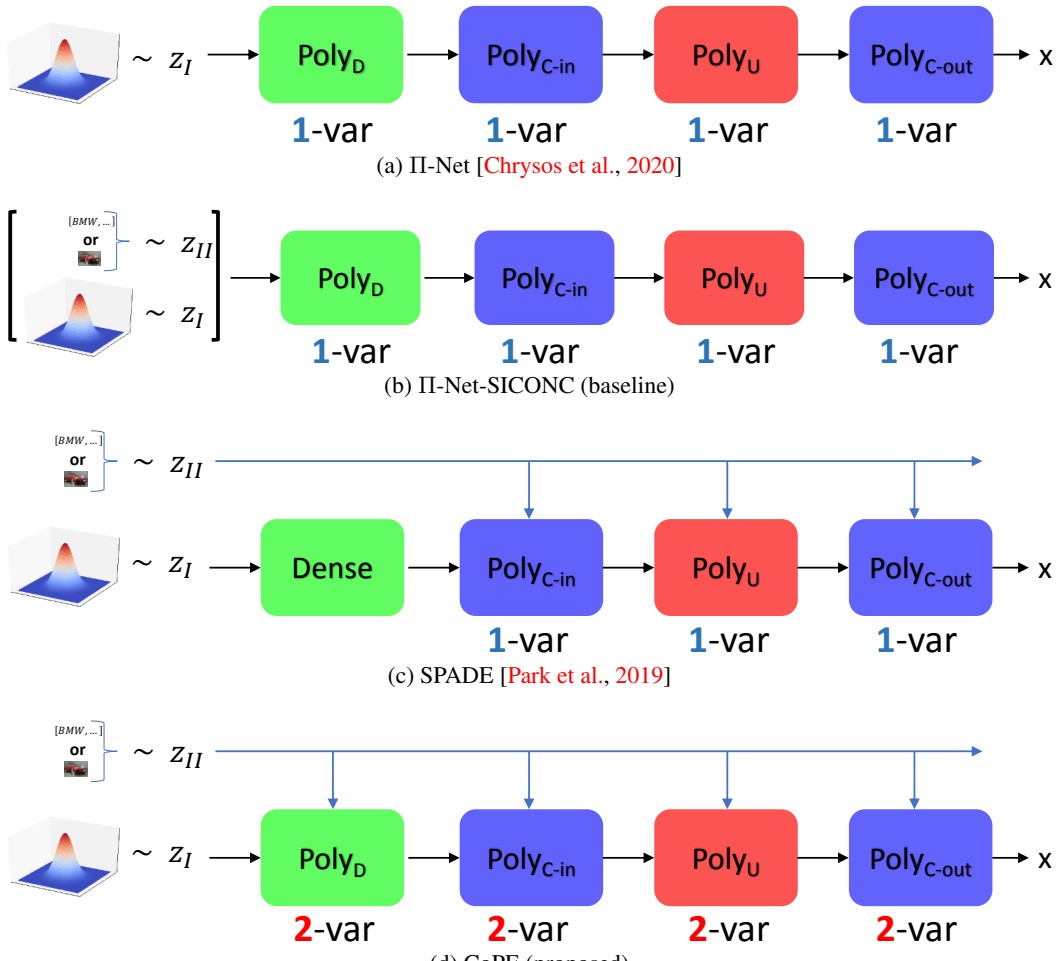

Figure 7: Abstract schematic of the different compared generators. All the generators are products of polynomials. Each colored box represents a different type of polynomial, i.e., the green box symbolizes polynomial(s) with dense layers, the blue box denotes convolutional or cross-correlation layers. The red box includes the up-sampling layers. (a) Π-Net implements a single-variable polynomial for modeling functions $x = G(z)$. Π-Net enables class-conditional generation by using conditional batch normalization (CBN). (b) An alternative to CBN is to concatenate the conditional variable in the input, as in Π-Net-SICONC. This also enables the non-discrete conditional variables (e.g., low-resolution images) to be concatenated. (c) SPADE implements a single-variable polynomial expansion with respect to the conditional variable $z_{II}$. This is substantially different from the polynomial with multiple-input variables, i.e., CoPE. Two additional differences are that (i) SPADE is motivated as a spatially-adaptive method (i.e., for continuous conditional variables), while CoPE can be used for diverse types of conditional variables, (ii) there is no polynomial in the dense layers in the SPADE. However, as illustrated in Π-Net converting the dense layers into a higher-order polynomial can further boost the performance. (d) The proposed generator structure.

the aforementioned sharing, the $N^{th}$ order expansion is described by:

$$\boldsymbol{x}_n = \left( \boldsymbol{A}_{[n,I]}^T \boldsymbol{z}_{\mathrm{I}} + \boldsymbol{A}_{[1,II]}^T \boldsymbol{z}_{\mathrm{II}} \right) * \left( \boldsymbol{V}_{[n]}^T \boldsymbol{x}_{n-1} + \boldsymbol{B}_{[n]}^T \boldsymbol{b}_{[n]} \right) \tag{20}$$

for $n = 2, \ldots, N$. Lastly, the factor $\boldsymbol{A}_{[1,II]}$ is a convolutional layer when the conditional variable is an image, while it is a fully-connected layer otherwise.

**Order of each polynomial in CoPE:** : Our experimental analysis demonstrates that there is not a single order of the polynomial expansion that works in all cases. The different polynomial expansions used (cf. Fig. 7) utilize different orders of expansion. The first set of polynomials with fully-connected

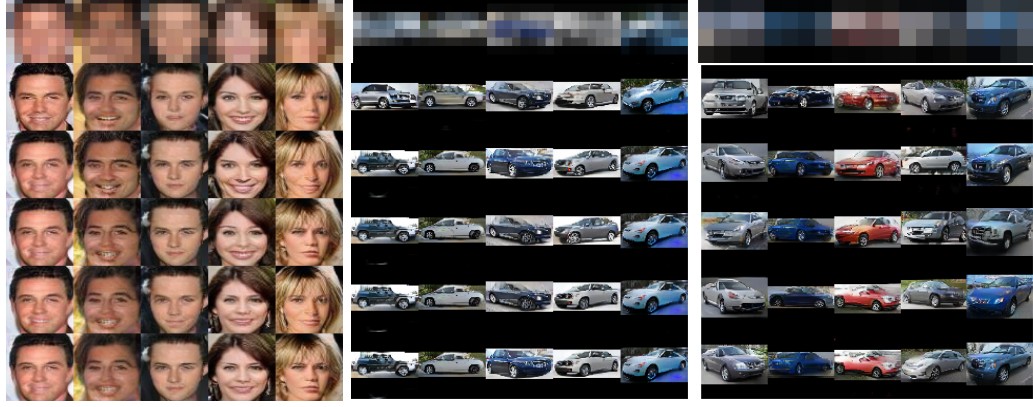

(a) Super-resolution $8\times$         (b) Super-resolution $8\times$         (c) Super-resolution $16\times$

Figure 8: Synthesized images for super-resolution by (a), (b) $8\times$, (c) $16\times$. The first row depicts the conditional input (i.e., low-resolution image). The rows 2-6 depict outputs of the CoPE when a noise vector is sampled per row. Notice how the noise changes (a) the smile or the pose of the head, (b) the color, car type or even the background, (c) the position of the car.

layers typically includes low-order polynomials (i.e., $1^{st} - 3^{rd}$ order), while the main polynomial includes $6^{th} - 9^{th}$ degree polynomial, and the output polynomial is also $1^{st} - 3^{rd}$ order polynomial.

# H   Additional experiments

Additional experiments and visualizations are provided in this section. Additional visualizations for class-conditional generation are provided in sec. H.1. An additional experiment with class-conditional generation with SVHN digits is performed in sec. H.2. An experiment that learns the translation of MNIST to SVHN digits is conducted in sec. H.3. To explore further the image-to-image translation, two additional experiments are conducted in sec. H.4. An attribute-guided generation is performed in sec. H.5 to illustrate the benefit of our framework with respect to multiple, discrete conditional inputs. This is further extended in sec. H.6, where an experiment with mixed conditional input is conducted. Finally, an additional diversity-inducing regularization term is used to assess whether it can further boost the diversity the synthesized images in sec. H.7.

## H.1   Additional visualizations in class-conditional generation

In Fig. 10 the qualitative results of the compared methods in class-conditional generation on CIFAR10 are shared. Both the generator of SNGAN and ours have activation functions in this experiment.

In Fig. 11 samples from the baseline Π-Net [Chrysos et al., 2020] and our method are depicted for the class-conditional generation on CIFAR10. The images have a substantial difference. Similarly, in Fig. 12 a visual comparison between Π-Net and CoPE is exhibited in Cars196 dataset. To our knowledge, no framework in the past has demonstrated such expressivity; CoPE synthesizes images that approximate the quality of synthesized images from networks with activation functions.

In Fig. 13, an inter-class interpolation of various compared methods in CIFAR10 are visualized. The illustrations of the intermediate images in SNGAN-CONC and SNGAN-ADD are either blurry or not realistic. On the contrary, in SPADE and CoPE the higher-order polynomial expansion results in more realistic intermediate images. Nevertheless, CoPE results in sharper shapes and images even in the intermediate results when compared to SPADE.

## H.2   Class-conditional generation on house digits

An experiment on class-conditional generation with SVHN is conducted below. SVHN images include (substantial) blur or other distortions, which insert noise in the distribution to be learned. In addition, some images contain contain a central digit (i.e., based on which the class is assigned), and

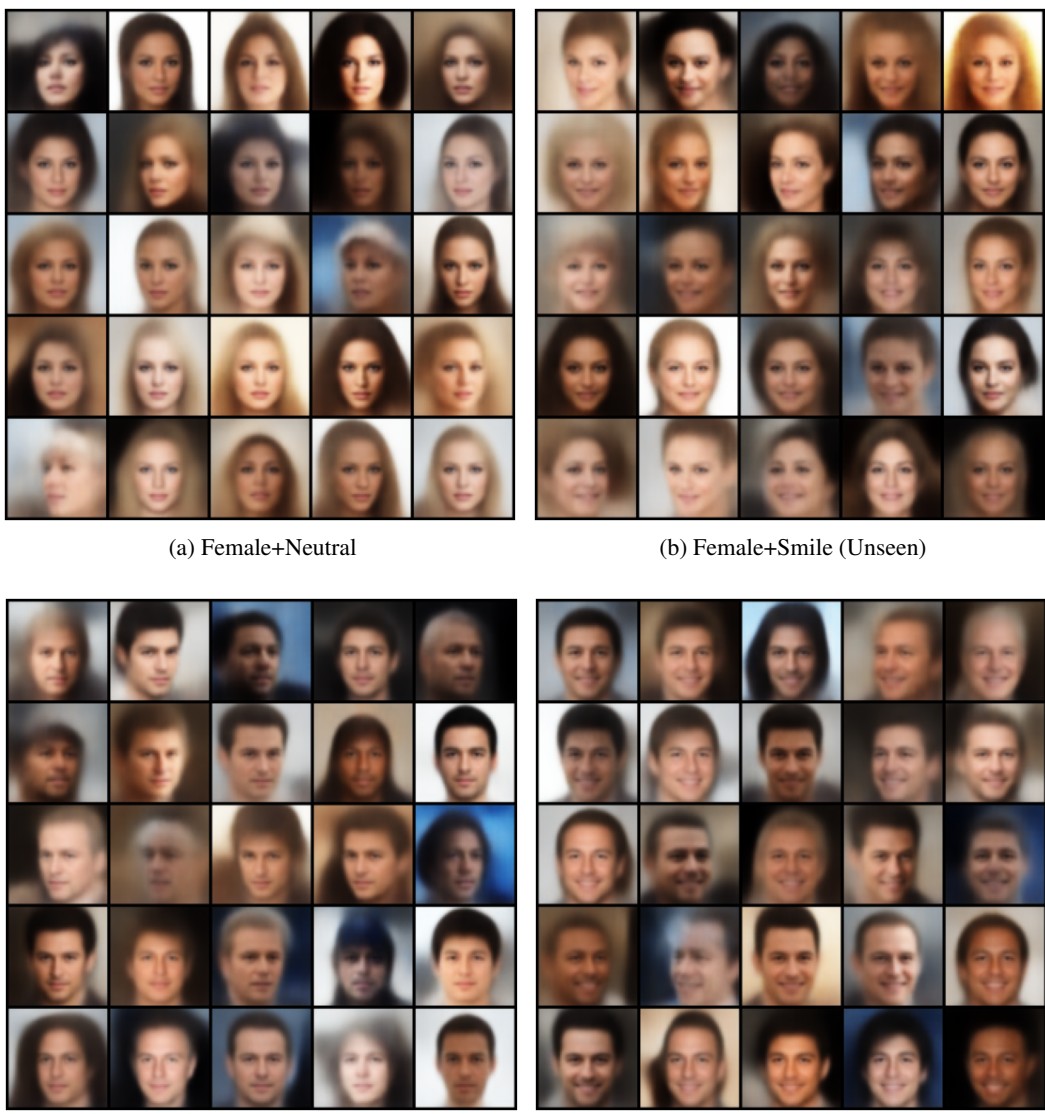

(a) Female+Neutral

(b) Female+Smile (Unseen)

(c) Male+Neutral

(d) Male+Smile

Figure 9: Additional synthesized images for CoPE-VAE. The setup of sec. 4.2 is used to provide the combinations, where the Woman+Smile is the combination not included in the training set. Notice that the proposed CoPE-VAE can synthesize images from all four combinations.

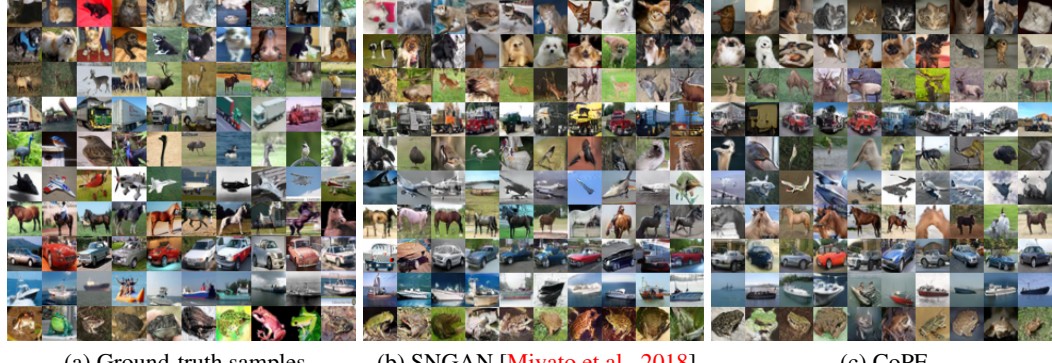

(a) Ground-truth samples      (b) SNGAN [Miyato et al., 2018]      (c) CoPE

Figure 10: Qualitative results on CIFAR10. Each row depicts random samples from a single class.

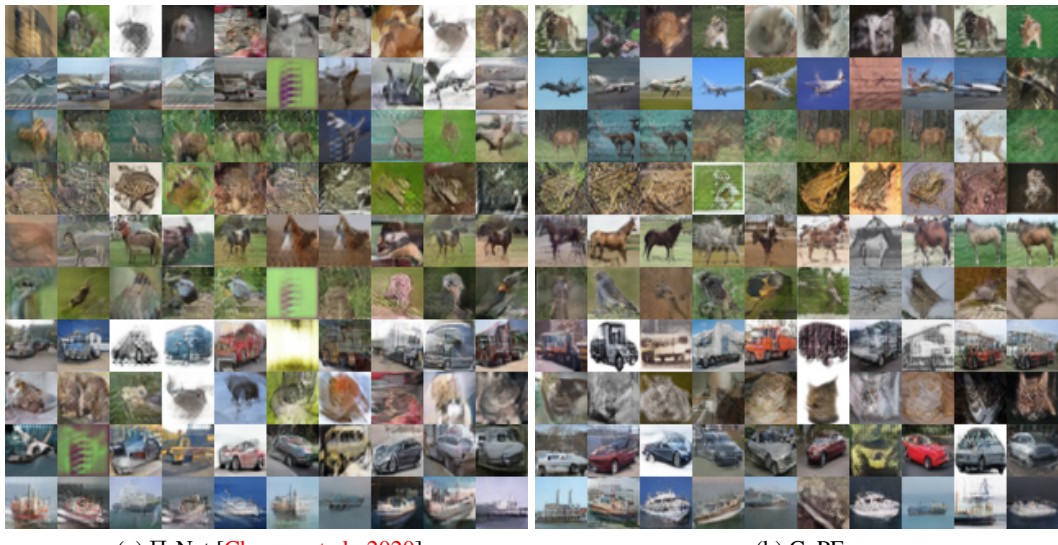

(a) Π-Net [Chrysos et al., 2020]      (b) CoPE

Figure 11: Qualitative results on CIFAR10 when the generator does not include activation functions between the layers. Each row depicts random samples from a single class; the same class is depicted in each pair of images. For instance, the last row corresponds to boats.

partial visibility of other digits. Therefore, the generation of digits of SVHN is challenging for a generator without activation functions between the layers.

Our framework, e.g., (14), does not include any activation functions. To verify the expressivity of our framework, we maintain the same setting for this experiment. Particularly, **the generator does not have activation functions between the layers**; there is only a hyperbolic tangent in the output space for normalization. The generator receives a noise sample and a class as input, i.e., it is a class-conditional polynomial generator.

The results in Fig. 14(b) illustrate that despite the noise, CoPE learns the distribution. As mentioned in the main paper, our formulation enables both inter-class and intra-class interpolations naturally. In the inter-class interpolation the noise $z_I$ is fixed, while the class $z_{II}$ is interpolated. In Fig. 14(d) several inter-class interpolations are visualized. The visualization exhibits that our framework is able to synthesize realistic images even with inter-class interpolations.

### H.3 Translation of MNIST digits to SVHN digits

An experiment on image translation from the domain of binary digits to house numbers is conducted below. The images of MNIST are used as the source domain (i.e., the conditional variable $z_{II}$), while

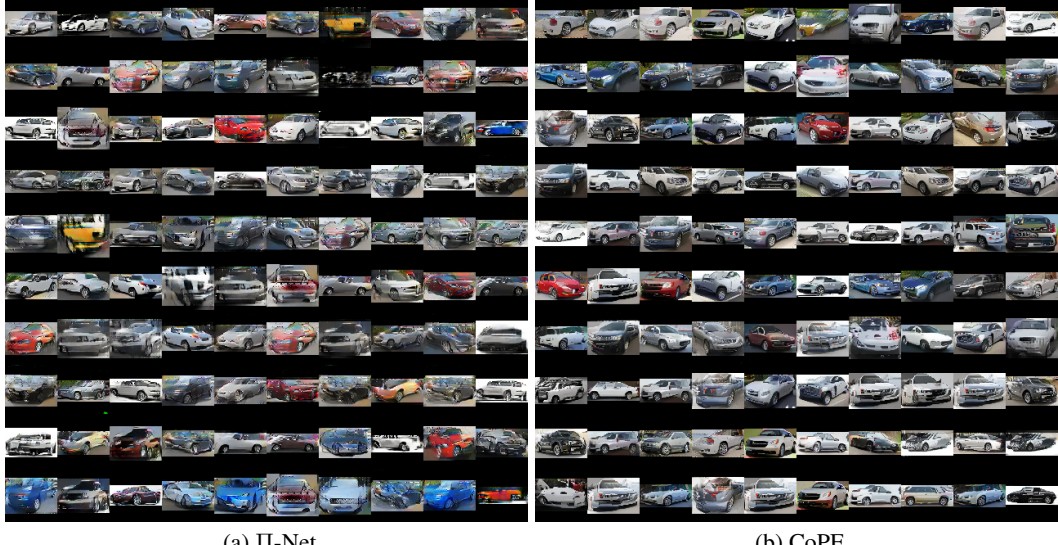

| (a) Π-Net | (b) CoPE |

Figure 12: Qualitative results on Cars196 when the generator does not include activation functions between the layers. Each row depicts cherry-picked samples from a single class; the same class is depicted in each pair of images. The differences between the synthesized images are dramatic.

the images of SVHN are used as the target domain. The correspondence of the source to the target domain is assumed to be many-to-many, i.e., each MNIST digit can synthesize multiple SVHN images. No additional loss is used, the setting of continuous conditional input from sec. 4.3 is used.

The images in Fig. 15 illustrate that CoPE can translate MNIST digits into SVHN digits. Additionally, for each source digit, there is a significant variation in the synthesized images.

### H.4 Translation of edges to images

An additional experiment on translation is conducted, where the source domain depicts edges and the target domain is the output image. Specifically, the tasks of edges-to-handbags (on Handbags dataset) and edges-to-shoes (on Shoes dataset) have been selected [Isola et al., 2017].

In this experiment, the MVP model of sec. 4.3 is utilized, i.e., a generator without activation functions between the layers. The training is conducted using *only* the adversarial loss. Visual results for both the case of edges-to-handbags and edges-to-shoes are depicted in Fig. 16. The first row depicts the conditional input $z_{\text{II}}$, i.e., an edge, while the rows 2-6 depict the synthesized images. Note that in both the case of handbags and shoes there is significant variation in the synthesized images, while they follow the edges provided as input.

### H.5 Multiple conditional inputs in attribute-guided generation

Frequently, more than one type of input conditional inputs are available. Our formulation can be extended beyond two input variables (sec. C); we experimentally verify this case. The task selected is attribute-guided generation trained on images of Anime characters. Each image is annotated with respect to the color of the eyes (6 combinations) and the color of the hair (7 combinations).

Since SPADE only accepts a single conditional variable, we should concatenate the two attributes in a single variable. We tried simply concatenating the attributes directly, but this did not work well. Instead, we can use the total number of combinations, which is the product of the individual attribute combinations, i.e., in our case the total number of combinations is 42. Obviously, this causes 'few' images to belong in each unique combination, i.e., there are 340 images on average that belong to each combination. On the contrary, there are 2380 images on average for each eye color.

SPADE and Π-Net are trained by using the two attributes in a single combination, while in our case, we consider the multiple conditional variable setting. In each case, only the generator differs

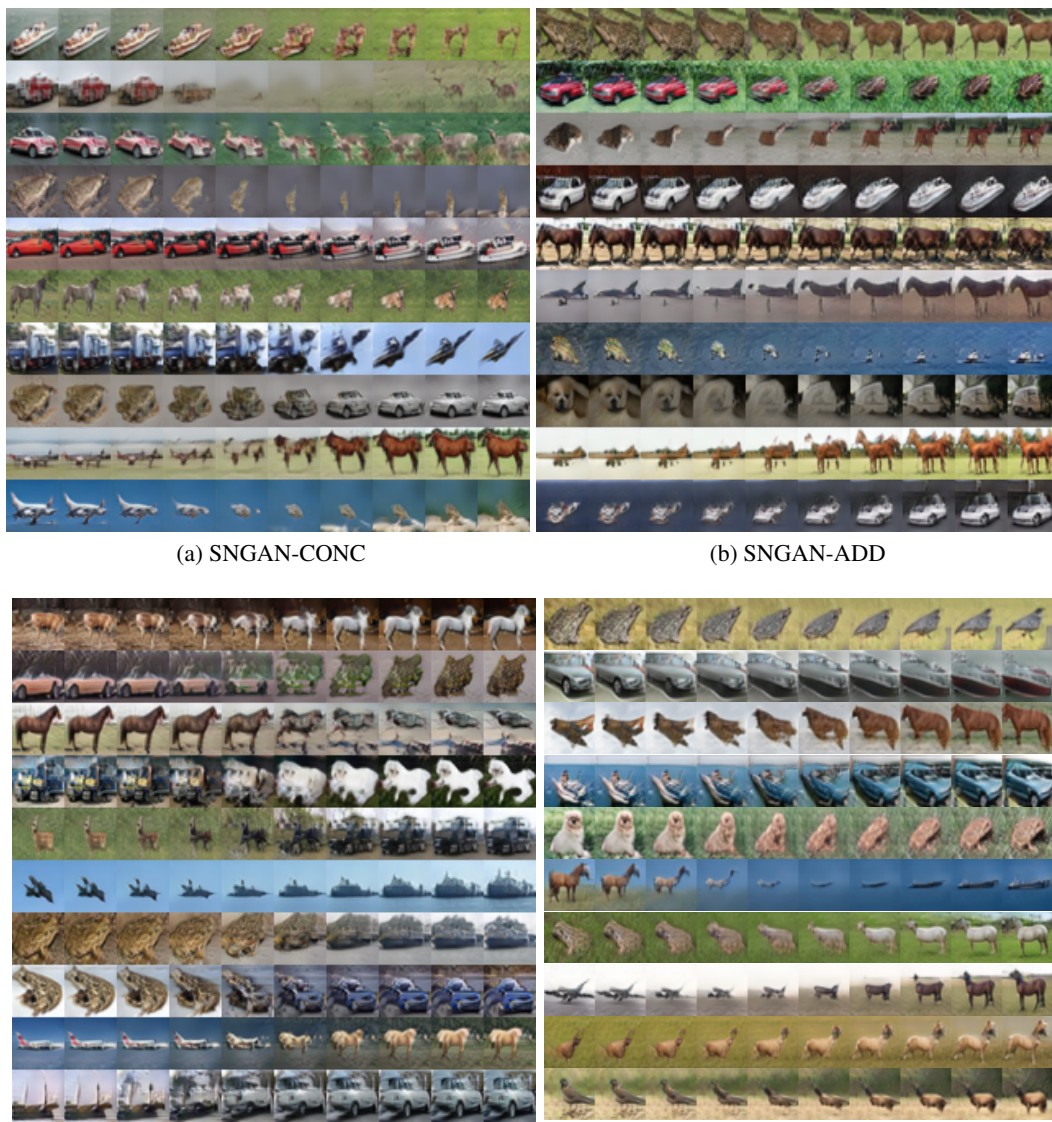

(a) SNGAN-CONC           (b) SNGAN-ADD

(c) SNGAN-SPADE           (d) SNGAN-CoPE

Figure 13: Inter-class linear interpolations across different methods. In inter-class interpolation, the class labels of the leftmost and rightmost images are one-hot vectors, while the rest are interpolated in-between; the resulting images are visualized. Many of the intermediate images in SNGAN-CONC and SNGAN-ADD are either blurry or not realistic. On the contrary, in SPADE and CoPE the higher-order polynomial expansion results in more realistic intermediate images. Nevertheless, CoPE results in sharper shapes and images even in the intermediate results when compared to SPADE.

depending on the compared method. In Fig. 17 few indicative images are visualized for each method; each row depicts a single combination of attributes, i.e., hair and eye color. Notice that SPADE results in a single image per combination, while in $\Pi$-Net-SINCONC there is considerable repetition in each case. The single image in SPADE can be explained by the lack of higher-order correlations with respect to the noise variable $z_1$.

In addition to the diversity of the images per combination, an image from every combination is visualized in Fig. 18. CoPE synthesizes more realistic images than the compared methods of $\Pi$-Net-SINCONC and SPADE.

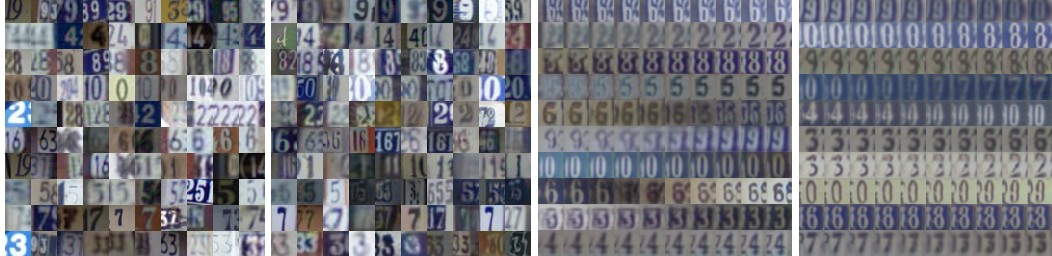

| (a) Ground-truth samples | (b) CoPE | (c) Intra-class interpolation | (d) Inter-class interpolation |

Figure 14: Synthesized images by CoPE in the class-conditional SVHN: (a) Ground-truth samples, (b) Random samples where each row depicts the same class, (c) Intra-class linear interpolation from a source (leftmost image) to the target (rightmost image), (d) inter-class linear interpolation. In inter-class interpolation, the class labels of the leftmost and rightmost images are one-hot vectors, while the rest are interpolated in-between; the resulting images are visualized. In all three cases ((b)-(d)), CoPE synthesizes realistic images.

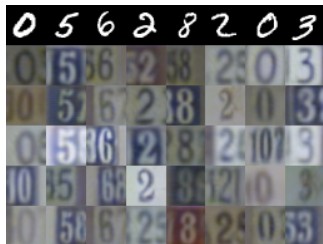

Figure 15: Qualitative results on MNIST-to-SVHN translation. The first row depicts the conditional input (i.e., a MNIST digit). The rows 2-6 depict outputs of the CoPE when a noise vector is sampled per row. Notice that for each source digit, there is a significant variation in the synthesized images.

## H.6 Multiple conditional inputs in class-conditional super-resolution

We extend the previous experiment with multiple conditional variables to the case of class-conditional super-resolution. The first conditional variable captures the class label, while the second conditional variable captures the low-resolution image.

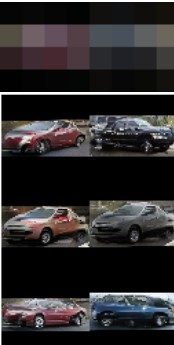

We use the experimental details of sec. 4.3 in super-resolution $8\times$. In Fig. 19, we visualize how for each low-resolution image the results differ depending on the randomly sampled class label. The FID in this case is 53.63, which is similar to the previous two cases. Class-conditional super-resolution (or similar tasks with multiple conditional inputs) can be of interest to the community and CoPE results in high-dimensional images with large variance.

## H.7 Improve diversity with regularization

As emphasized in sec. I, various methods have been utilized for synthesizing more diverse images in conditional image generation tasks. A reasonable

Figure 19: Three-variable input generative model.

question is whether our method can be used in conjunction with such methods, since it already synthesizes diverse results. Our hypothesis is that when CoPE is used in conjunction with any diversity-inducing technique, it will further improve the diversity of the synthesized images. To assess the hypothesis, we conduct an experiment on edges to images that is a popular benchmark in such diverse generation tasks [Zhu et al., 2017b, Yang et al., 2019].

The plug-n-play regularization term of Yang et al. [2019] is selected and added to the GAN loss during the training. The objective of the regularization term $\mathcal{L}_{reg}$ is to maximize the following term:

$$\mathcal{L}_{reg} = \min\left(\frac{||G(\boldsymbol{z}_{I,1}, \boldsymbol{z}_{II}) - G(\boldsymbol{z}_{I,2}, \boldsymbol{z}_{II})||_1}{||\boldsymbol{z}_{I,1} - \boldsymbol{z}_{I,2}||_1}, \tau\right) \qquad (21)$$

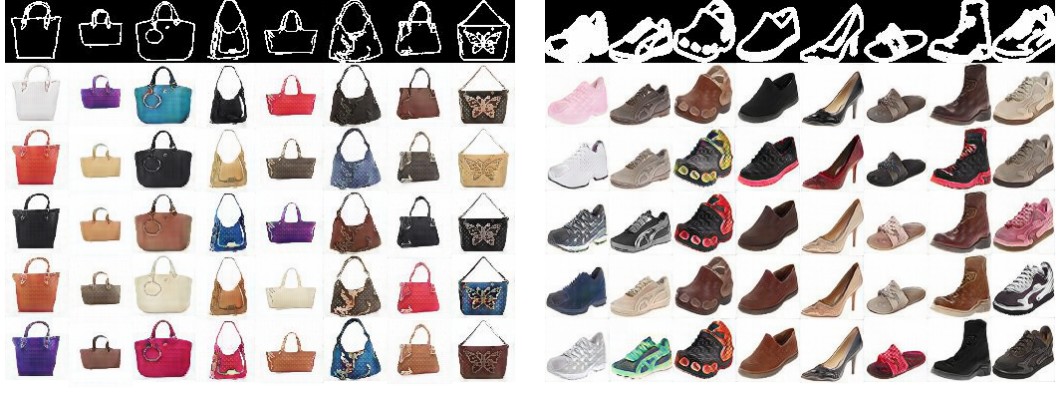

(a) edges-to-handbags           (b) edges-to-shoes

Figure 16: Qualitative results on edges-to-image translation. The first row depicts the conditional input (i.e., the edges). The rows 2-6 depict outputs of the CoPE when we vary $z_I$. Notice that for each edge, there is a significant variation in the synthesized images.

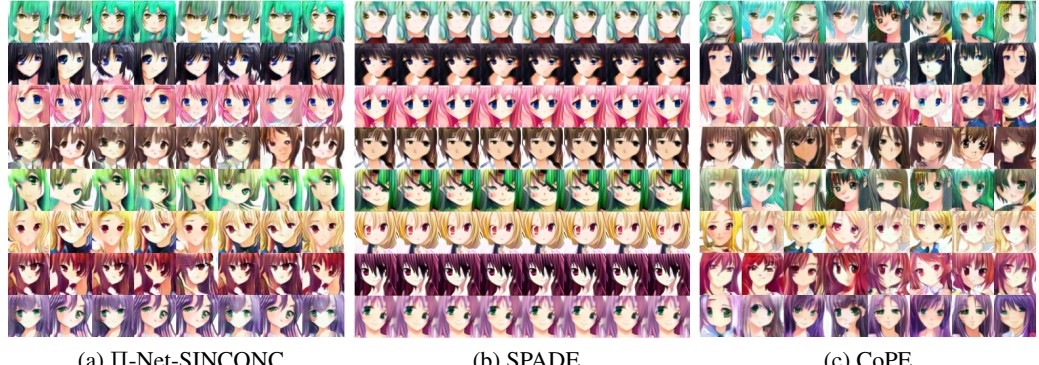

(a) Π-Net-SINCONC        (b) SPADE        (c) CoPE

Figure 17: Each row depicts a single combination of attributes, i.e., hair and eye color. Please zoom-in to check the finer details. The method of SPADE synthesizes a single image per combination. Π-Net-SINCONC synthesizes few images, but not has many repeated elements, while some combinations result in unrealistic faces, e.g., the $5^{th}$ or the $7^{th}$ row. On the contrary, CoPE synthesizes much more diverse images for every combination.

where $\tau$ is a predefined constant, $z_{I,1}, z_{I,2}$ are different noise samples. The motivation behind this term lies in encouraging the generator to produce outputs that differ when the input noise samples differ. In our experiments, we follow the implementation of the original paper with $\tau = 10$.

The regularization loss of (21) is added to the GAN loss; the architecture of the generator remains similar to sec. H.4. The translation task is edges-to-handbags (on Handbags dataset) and edges-to-shoes (on Shoes dataset). In Fig. 20 the synthesized images are depicted. The regularization loss causes more diverse images to be synthesized (i.e., when compared to the visualization of Fig. 16 that was trained using only the adversarial loss). For instance, in both the shoes and the handbags, new shades of blue are now synthesized, while yellow handbags can now be synthesized.

The empirical results validate the hypothesis that our model can be used in conjunction with diversity regularization losses in order to improve the results. Nevertheless, the experiment in sec. H.4 indicates that a regularization term is not necessary to synthesize images that do not ignore the noise as feed-forward generators had previously.

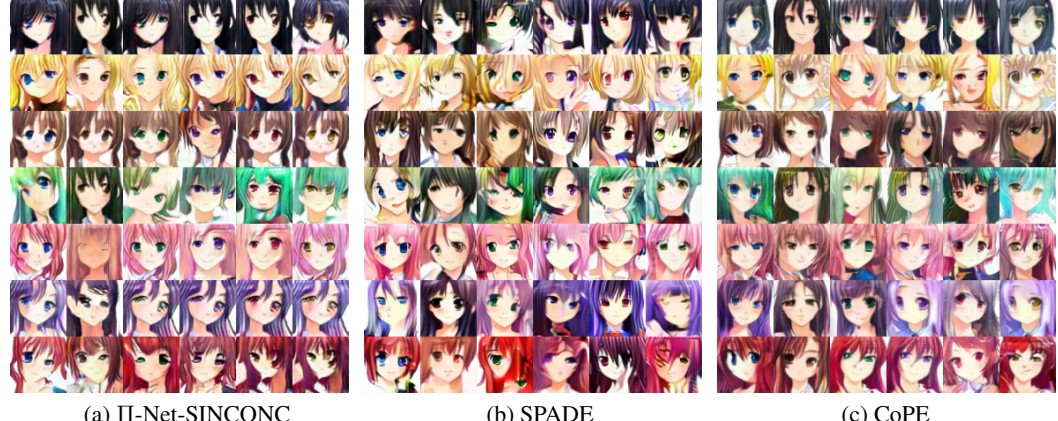

|                    |                 |              |
| :---:              | :---:           | :---:        |
| (a) Π-Net-SINCONC  | (b) SPADE       | (c) CoPE     |

Figure 18: Each row depicts a single hair color, while each column depicts a single eye color. SPADE results in some combinations that violate the geometric structure of the face, e.g., $3^{rd}$ column in the last row. Similarly, in Π-Net-SINCONC some of the synthesized images are unrealistic, e.g., penultimate row. In both SPADE and Π-Net-SINCONC, in some cases the eyes do not have the same color. CoPE synthesizes images that resemble faces for every combination.

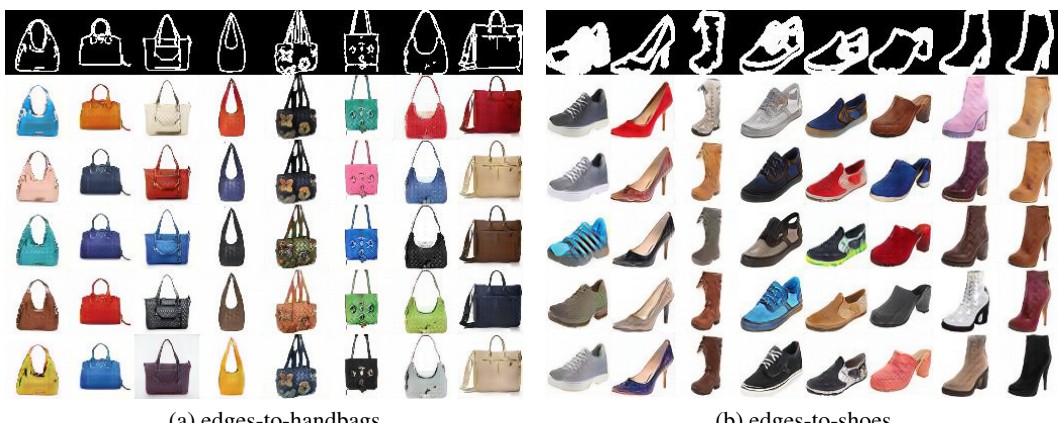

|                        |                     |
| :---:                  | :---:               |
| (a) edges-to-handbags  | (b) edges-to-shoes  |

Figure 20: Qualitative results on edges-to-image translation with regularization loss for diverse generation (sec. H.7). The first row depicts the conditional input (i.e., the edges). The rows 2-6 depict outputs of the CoPE when we vary $z_I$. Diverse images are synthesized for each edge. The regularization loss results in 'new' shades of blue to emerge in the synthesized images in both the shoes and the handbags cases.

## H.8 Generation of unseen attribute combinations

In this section, we highlight how the proposed CoPE-VAE compares to the other baselines for the task of generating unseen attribute combinations. Following the benchmark in [Georgopoulos et al., 2020] we compare to cVAE [Sohn et al., 2015], VampPrior [Tomczak and Welling, 2017] and MLC-VAE-CP/T [Georgopoulos et al., 2020]. The results in Figure 4 and Table 6 showcase the efficacy of our method in recovering the unseen attribute combination of ("smiling", "female"). In particular, the quantitative comparison in Table 6 shows that our method is on par or outperforms the baseline methods in both attribute synthesis and attribute transfer. The quantitative results were obtained using attribute classifiers trained on CelebA for gender and smile.

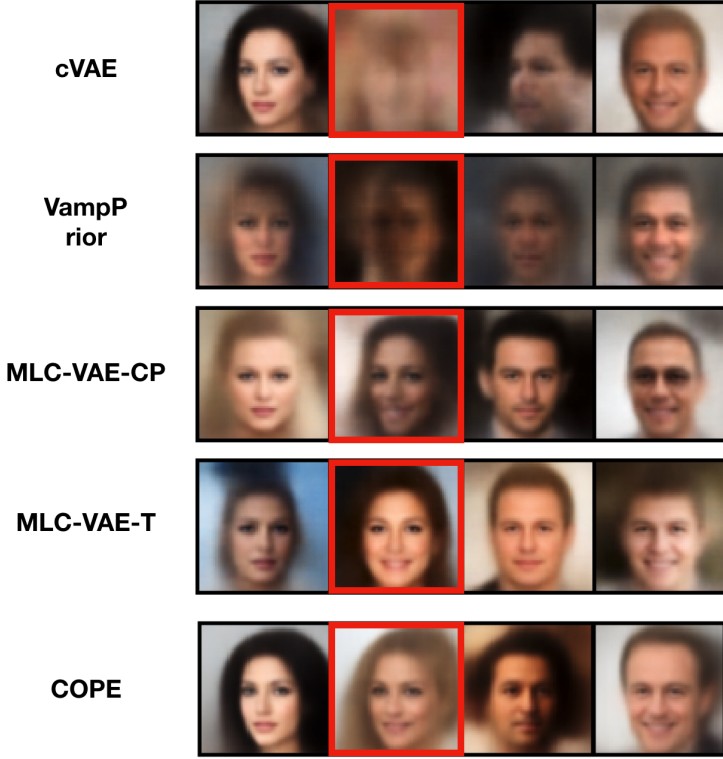

Figure 21: Qualitative comparison on CelebA. The missing combination, i.e., ("Smiling", "Female"), is in the red rectangle. Results for cVAE, VampPrior, MLC-VAE-CP and MLC-VAE-T are taken directly from [Georgopoulos et al., 2020].

| Model | Attribute synthesis | | Attribute transfer | |
|---|---|---|---|---|
| | Acc. Gender (%) ↑ | Acc. Smile (%) ↑ | Acc. Gender (%) ↑ | Acc. Smile (%) ↑ |
| cVAE | 18 | 7.6 | 23.1 | 9 |
| cVampPrior | 17.1 | 2.8 | 44.3 | 5.8 |
| MLC-VAE-CP | 96.4 | 94 | 95 | 90.6 |
| MLC-VAE-T | 99.4 | 93.5 | 99.4 | 91.5 |
| CoPE-VAE | 99.9 | 92.6 | 99.5 | 92.38 |

Table 6: Quantitative comparison on CelebA using attribute classifiers. The reported results are **only for the unseen attribute combination**, i.e., ("Smiling", "Female"). Results for cVAE, VampPrior, MLC-VAE-CP and MLC-VAE-T are taken directly from [Georgopoulos et al., 2020].

# I  Diverse generation techniques and its relationship with CoPE

One challenge that often arises in conditional data generation is that one of the variables gets ignored by the generator [Isola et al., 2017]. This has been widely acknowledged in the literature, e.g., Zhu et al. [2017b] advocates that it is hard to utilize a simple architecture, like Isola et al. [2017], with noise. A similar conclusion is drawn in InfoGAN [Chen et al., 2016] where the authors explicitly mention that additional losses are required, otherwise the generator is 'free to ignore' the additional variables. To mitigate this, a variety of methods have been developed. We summarize the most prominent methods from the literature, starting from image-to-image translation methods:

- BicycleGAN [Zhu et al., 2017b] proposes a framework that can synthesize diverse images in image-to-image translation. The framework contains 2 encoders, 1 decoder and 2 discriminators. This results in multiple loss terms (e.g., eq.9 of the paper). Interestingly, the authors utilize a separate training scheme for the encoder-decoder and the second encoder as training together 'hides the information of the latent code without learning meaningful modes'.

- Almahairi et al. [2018] augment the deterministic mapping of CycleGAN [Zhu et al., 2017a] with a marginal matching loss. The framework learns diverse mappings utilizing the additional encoders. The framework includes 4 encoders, 2 decoders and 2 discriminators.

- MUNIT [Huang et al., 2018] focuses on diverse generation in unpaired image-to-image translation. MUNIT demonstrates impressive translation results, while the inverse translation is also learnt simultaneously. That is, in case of edges-to-shoes, the translation shoes-to-edges is also learnt during the training. The mapping learnt comes at the cost of multiple network modules. Particularly, MUNIT includes 2 encoders, 2 decoders, 2 discriminators for learning. This also results in multiple loss terms (e.g., eq.5 of the paper) along with additional hyper-parameters and network parameters.

- Drit++ [Lee et al., 2020] extends unpaired image-to-image translation with disentangled representation learning, while they allow multi-domain image-to-image translations. Drit++ uses 4 encoders, 2 decoders, 2 discriminators for learning. Similarly to the previous methods, this results in multiple loss terms (e.g., eq.6-7 of the paper) and additional hyper-parameters.

- Choi et al. [2020] introduce a method that supports multiple target domains. The method includes four modules: a generator, a mapping network, a style encoder and a discriminator. All modules (apart from the generator) include domain-specific sub-networks in case of multiple target domains. To ensure diverse generation, Choi et al. [2020] utilize a regularization loss (i.e., eq. 3 of the paper), while their final objective consists of multiple loss terms.

The aforementioned frameworks contain additional network modules for training, which also results in additional hyper-parameters in the loss-function and the network architecture. Furthermore, the frameworks focus exclusively on image-to-image translation and not all conditional generation cases, e.g., they do not tackle class-conditional or attribute-based generation.

Using regularization terms in the loss function has been an alternative way to achieve diverse generation. Mao et al. [2019], Yang et al. [2019] propose simple regularization terms that can be plugged into any architecture to encourage diverse generation. Lee et al. [2019] propose two variants of a regularization term, with the 'more stable variant' requiring additional network modules.

Even though our goal is not to propose a framework for diverse generation, the synthesized images of CoPE contain variation when the noise is changed. However, more diverse results can be exhibited through a dedicated diverse generation technique. We emphasize that our method can be used in conjunction with many of the aforementioned techniques to obtain more diverse examples. We demonstrate that this is possible in an experiment in sec. H.7.