# OpenReview forum: "Conditional Generation Using Polynomial Expansions"
_NeurIPS.cc/2021/Conference — NeurIPS 2021 Poster_

### Official Review · Reviewer_Mncu · 2021-07-12

**Rating:** 4
**Confidence:** 4

**Summary:**

## Paper Summary:

Deep Polynomial Neural Networks [Chrysos et al 2020] introduce layers that model higher-order polynomial interactions of a single input. This paper extends PNN’s to the setting where atleast two inputs are provided. (For eg, in conditional image generation, the inputs are the noise variable (z) and class variable ©).  In particular, they incorporate an additional learnable dense projection to incorporate this into the initial PNN formulation (Eq 2)

The authors provide experimental results on a number of benchmarks to showcase the applicability of their technique.


**Limitations And Societal Impact:**

Yes

**Main Review:**

## Pros

* Generative modeling to incorporate multiple auxiliary variables is an interesting direction to pursue.
* I appreciate the authors efforts to showcase the applicability of their method to a number of benchmarks.

## Cons:

The paper is not well structured (for eg, self-contained with all the background material) and the experiments are weak. Even if the proposed architectural change is incremental (wrt PNN's), consistent and strong improvements over multiple benchmarks with solid experiments would attain a more favourable rating,
Here are my suggestions for potential improvements.

### Structure:
* The paper highlights in the introduction that a trivial modification would be to concatenate the two inputs. However, the modification proposed amounts to tiling the noise input across the image and then applying a dense projection which is arguably almost as trivial.
* A majority of the methods section is the contribution of [Chrysos et al 2020]. For eg, the recursive formulation, product of polynomials and the normalization stability tricks. I suggest that the methods section should solely focus on how [Chrysos et al 2020] is modified to incorporate multiple variables. The prerequisites have to be either moved to a background section or the methods section should be rewritten to make this distinction.

### Experiments:
* What are N (order) and K (projection dim) used in the experiments? Are they fixed across all tasks and all baselines?
* [Karras et al 2020] discuss the 3 variants used in their paper, i.e CCP, NCP and NCP-skip. Why was only the CPP-variant used?
* How was SPADE modified to incorporate conditioning?
* SPADE is the only baseline, while `GAN-CONC, GAN-ADD and maybe Π-Net-SICONC` are ablations of the current technique. The tables could be changed to reflect this.
* Some bolded results may not be significant.
  * Table 3: On IS, SNGAN-CoPE and SNGAN-Spade are within standard deviation
  * Table 3: On FID, SNGAN-CoPe and SNGAN are comparable.
  * Table 4: On CIFAR 10, CoPe is comparable to pi-net.
  *  Table 5: On CARS196, CoPE is comparable to SPADE-CoPE.
* Section 4.2 is qualitative and hard to judge because of no baselines. I suggest to move it to the appendix.
* In Table 5, Baseline SNGAN is missing. Why is SPADE-CoPE introduced here but not in the other tables? Why are there two modifications of SPADE? The “CONC” and “ADD” variants seem to be missing as well.
* The paper mentions StyleGAN in the introduction, but there are no StyleGAN variants (for eg, style + label) in the experiments.

Minor Comments:
* Line 20: PNNs have demonstrated impressive generation results in faces, animals, cars [Karras et al., 2020b], 22 paintings, medical images [Karras et al., 2020a]:
C: I suggest the citation be changed to Chrysos et al. [2019]. StyleGAN is a specific type of PNN + other innovations. StyleGan itself has been applied to image-to-image in StarGanv2 (that the authors cite)
* Line 22: paintings, images -> paintings and images.
* Line 34 - for conditional data generation -> “for conditional data generation that employs or uses PNNs”
* Line 39 and Line 44: In CoPE, different architectures can be defined simply by changing the recursive formulation.
	It is not clear what “different architectures” mean. I do not believe the “formulation” is really changed since Eq 2 is applied everywhere
* Line 152 - Line 155: Is this used in any experiments? In any case, this can be easily incorporated in equation 2.
* Line 156 - Line 165. The description of the product of polynomials method is confusing. I found the description in Section 3.2 and Figure 5 of [Karras et al 2020] much easier and intuitive to follow.
* Line 170 -> What does “representations” mean here? Is batch-normalization applied at the output of Eq 4?


**Time Spent Reviewing:**

6 - 7 hours

---

> ### Author Response · Authors · 2021-08-09
> **Response to the reviewer Mncu**
>
>
> We thank the reviewer for the time and effort to provide a thorough review. We address their concerns below:
>
>
> > What are N (order) and K (projection dim) used in the experiments? Are they fixed across all tasks and all baselines?
>
> * Order of CoPE: We have used a different order for the different polynomials in CoPE. As we demonstrate in Fig. 7(d) (supplementary) there are a number of polynomials used in CoPE. The first set of polynomials with fully-connected layers typically include low-degree polynomials (i.e., 1-3 degree), while the main polynomial includes 6-9 degree polynomial, and the output polynomial is also 1-3 degree polynomial. We believe that our open source code will allow the interested practitioner to understand all the details.
>
> * Projection dimension $K$: Similarly to the order of each polynomial, the projection dimension differs per polynomial. Indicatively, in the SNGAN-based generator, $K=128$.
> ________
>
> > [Karras et al 2020] discuss the 3 variants used in their paper, i.e CCP, NCP and NCP-skip. Why was only the CCP-variant used?
>
> We assume that the reviewer refers to the variants proposed in [Chrysos et al 2020].
>
> Indeed, the model proposed in the main paper can be casted as a CCP-variant. However, an additional model that is an NCP-variant is proposed in the supplementary (equation 14). More models (even beyond the original three variants of [Chrysos et al 2020]) can be proposed by changing the assumptions of the tensor decomposition. In fact, we believe it is an interesting avenue for task-specific architectures to be proposed and we will discuss it in the camera-ready version of the paper.
>
> ________
>
> > How was SPADE modified to incorporate conditioning?
>
> SPADE was originally proposed using semantic labels in the form of an image as the conditional variables for conditional generation. Instead, we use the classes (in sec. 4.1) or the low-resolution image (in sec. 4.3) as the conditional variable.
>
> ________
>
> > Some results might not be significant improvements over the baselines.
>
> Our methodology demonstrates how CoPE generalizes frameworks such as SPADE. Then, we construct SPADE-CoPE to illustrate how we can improve SPADE. Indeed, some improvements might be more significant than others. The main contribution of this work is a general framework for the construction of networks of multiple inputs, the efficacy of which is demonstrated by a  wide range of experiments across tasks.
>
> ________
>
> > In Table 5, Baseline SNGAN is missing. Why is SPADE-CoPE introduced here but not in the other tables? Why are there two modifications of SPADE? The “CONC” and “ADD” variants seem to be missing as well.
>
> * In Table 5 we compare the polynomial networks with no activation functions (e.g., Π-net, SPADE). Therefore, it would be unfair to include SNGAN as a baseline.
> * SPADE-CoPE: Our preliminary experiments demonstrated that SPADE-CoPE had a minor difference to SPADE in SNGAN-based generators. Since this is not the main proposed model, we do not include it in the main experiment.
> * There is only one modification to SPADE, i.e., SPADE-CoPE. The SPADE is the original model following the instructions of the authors. As explained in the related work and the method, SPADE is a polynomial with respect to the conditional variable, so we include it as a baseline. Then, we demonstrate how to improve it to SPADE-CoPE to bring it closer to CoPE.
> * There is some explanation on lines 727-759 in the supplementary (sec. G) of GAN-CONC and GAN-ADD. Succinctly:  GAN-CONC is a GAN-variant that concatenates the two input variables. GAN-ADD is a GAN-variant that adds the embeddings of the two input variables.
> ________
>
> > The paper mentions StyleGAN in the introduction, but there are no StyleGAN variants (for eg, style + label) in the experiments.
>
> StyleGAN has been casted as a polynomial network before (e.g. [Chrysos et al 2020] demonstrate that the synthesis network of the generator is a special case of the NCP variant) and they improve upon it by extending the mapping network into a polynomial network. To that end, we use the improved version of Π-nets in our experiments.
>
> ________
>
> > Line 39 and Line 44: In CoPE, different architectures can be defined simply by changing the recursive formulation. It is not clear what “different architectures” mean. I do not believe the “formulation” is really changed since Eq 2 is applied everywhere
>
> The selected tensor factorization can result in different recursive formulations that in turn construct a different network architecture. For instance, Figure 1 and Figure 6 demonstrate two variants; notice that the V_[n] and the B_[n] b_[n] do not exist in Figure 1.
>
> ________
>
> > Line 152 - Line 155: Is this used in any experiments? In any case, this can be easily incorporated in equation 2.
>
> Yes, it is used in sec. H.5 and H.6 (lines 1010-1041 in the supplementary); we agree with the reviewer that this can be easily included in equation 2.

---

### Official Review · Reviewer_La4a · 2021-07-14

**Rating:** 7
**Confidence:** 3

**Summary:**

In this paper, the author proposes a new framework to extend deep polynomial neural networks (PNNs) to multi-variable polynomial neural networks. The proposed PNN framework accepts multiple variables, thus enabling conditional generation where at least two variables, noise and conditioned, are needed. Extensive experiments are conducted to demonstrate the feasibility of conditional PNN under the framework.

**Limitations And Societal Impact:**

The authors adequately addressed the limitations and potential negative societal impact of their work.

**Main Review:**

This paper studies extending deep polynomial neural networks (PNNs) to multi-variable polynomial neural networks. The author argues that the although PNN or PNN-like net can (at least partially) explain the success of current state-of-the-art GANs, current framework either only accepts single variable and thus are only enough for unconditional generation where only noise (i.e. latent vector), or needs special treatment of the conditions that are tailored to a particular tasks.

Thus the author proposed to extend PNNs, such that PNNs in a new framework are framed as polynomials of multiple variables (Sec 3.1). Details are taken care of in Sec 3.2 to for empericaly efficiency and stability. The proposed extension looks new in the context of PNN and sets itself from other PNN works in (1) introducing extra variables and deriving formulas  connecting multiple variables in polynomial networks. and (2) providing a universal way to represent conditions that could be either continuous or discrete.

Conducted experiments look pretty solid and comprehensive. The authors cover diverse conditional generations tasks where consciousness can be either delicate classes/attributes or continuous images. As this work is to present a new framework that works for a wide range of tasks, it's ok that the qualitative results are not beating the vis-a-vis SOTA model in particular tasks.

Also this paper is mostly well written and easy to follow. details are well covered in the appendix. Overall, I think this solid work deserves a clear acceptance.

Nitpicking: Typo on  L78 "Alnother"


**Time Spent Reviewing:**

2 hours

---

> ### Author Response · Authors · 2021-08-09
> **Response to the reviewer La4a**
>
> We are thankful to the reviewer for the appreciation of our work overall. We appreciate the thorough reading; the typo will be fixed in the final version of the paper.

---

### Official Review · Reviewer_oDAB · 2021-07-17

**Rating:** 5
**Confidence:** 4

**Summary:**

The paper has shown an improvisation on the Pi-Net, deep polynomial neural networks, on a conditional multiplicative setting that is shown to work on different computer vision tasks such as conditional image generation, inverse tasks like translation, inpainting, etc. The paper has shown experimental results improvement on class-conditional, image-conditional, and mixed conditional scenarios.

**Ethics Review Area:**

["I don’t know"]

**Limitations And Societal Impact:**

 The authors have adequately addressed the limitations and potential negative societal impact of their work

**Main Review:**

Strengths
----------------------

1. The paper has shown an organic way to condition the Pi-Net. The methodology is shown to perform well in various alternate computer vision tasks.
2. The methods section is well-written and the intuition provided to readers who are not familiar with the concept of polynomial networks is very useful. I really liked the detailed discussion in Sup. Mat.
3. The figures are succinct and informative, giving a clear picture of the points the authors want to illustrate. I appreciate the multiple evaluations with mean and standard deviation reported.


Weakness
----------------------

+++++++++++++++++++++

1. Significance of the Study:

1.1. The paper has shown a comparison with SPADE, StyleGAN, sBN methods. But, none of them is a state-of-the-art method (if I follow the "Does it advance the state of the art in a demonstrable way?" of the reviewer guideline). The choice of SPADE for the inverse problem and conditional generation is not very appropriate when there are more recent conditional generations and inverse problem-related wors such as [1], [2], etc.

1.2. The reviewer is not convinced that cBN/sBN is not applicable to continuous conditions, as sBN predicts the BN parameters from the continuous latent variable. Clarification on why the paper has given such information is much needed.

++++++++++++++++++++

2. Experiments:

2.1.  Discussion on Quantitative Result is missing: The experiments are all done at 64x64 compared to other SOTAs, like BigGAN 512 X 512 on ImageNet. Any justification why this is so?

2.2.   Moreover, the improvement on a commonly used architecture (ResNet) is minimal. It will be nice if the paper provides theoretical justification why the method does not perform well under common settings. (if I follow the "Does it advance the state of the art in a demonstrable way?" of the reviewer guideline)

2.3.  Score Disparity: I am a bit confused on why the paper has shown the FID of SNGAN 21.7? Any reason for that? I tried the original code but failed to reproduce the FID of the SNGAN [3] shown in the paper. Which source code the paper has tried? What is the seed value, what are the learning rate, batch size, and other hyperparameter values?

2.4. Time complexity: How about the number of parameters or sampling speed? Is the comparison in the qualitative results table fair in terms of complexity?

2.5. What is the seed value we have taken for this experiment? Are those results are shown for one seed or an interval? If, interval what was the distribution that we sampled seeds?

2.6. [Minor] The paper can move the MNIST-SVHN, Edge-Image translation results to the main draft.

3. Novelty: Using the conditional on an unconditional Pi-Net seems not a good novelty. What architectural changes happen here instead of adding a new path for the conditional? Why the setup works well than a GAN? Is there any theoretical evidence in the paper? Am I missing something?


Ref
-------------------
[1]  Image2StyleGAN++: How to Edit the Embedded Images?

[2]  GAN Compression: Efficient Architectures for Interactive Conditional GANs

[3] Spectral normalization for generative adversarial networks

**Time Spent Reviewing:**

10hrs

---

> ### Author Response · Authors · 2021-08-09
> **Response to the reviewer oDAB**
>
> We thank the reviewer for the time and effort to provide a thorough review. We address their concerns below:
>
> >  The experiments are all done at 64x64 compared to other SOTAs, like BigGAN 512 X 512 on ImageNet. Any justification why this is so?
>
> Training BigGAN requires **at least 8 GPUs for a single training** (as per the authors). Unfortunately, we do not have access to resources at this scale and so we cannot obtain results in this resolution. Instead, we show both methodologically and with reasonable experiments (on a single GPU) the benefits of CoPE; these experiments i) have a much **smaller energy footprint** (than BigGAN) and ii) **can be replicated** by all practitioners (e.g., with Google Collab).
>
> ________
>
> > I am a bit confused on why the paper has shown the FID of SNGAN 21.7? Any reason for that? I tried the original code but failed to reproduce the FID of the SNGAN [3] shown in the paper. Which source code the paper has tried? What is the seed value, what are the learning rate, batch size, and other hyperparameter values?
>
> We **do not** report FID of 21.7; the *reported FID is for SNGAN is 14.70* (table 3). The code we have used is the publicly available implementation of the authors or SNGAN, i.e., https://github.com/pfnet-research/sngan_projection . We have used the conditional batch normalization setup described in their works for conditional training; the rest of the parameters remain as in the original code.
>
> ________
>
> > How about the number of parameters or sampling speed? Is the comparison in the qualitative results table fair in terms of complexity?
>
> We welcome the suggestion of the reviewer. We measure the sampling speed/parameters for both the SNGAN-based experiment and for the Π-net-based experiment in class-conditional generation.
>
> The parameters (indicated in millions) along with the speed of generating images (run in a single GPU with batch size 32 and repeated 100 to report the mean) are reported below.
>
> The table below compares the parameters and the speed for the SNGAN-based comparisons:
>
> |    Method            |Parameters (million) |Speed (in ms)                         |
> |---------------------|-----------------|-----------------------|
> |SNGAN        |4.3|17.57|
> |SNGAN-CONC    |5.3|21.69|
> |SNGAN-ADD    |4.8|19.52|
> |SNGAN-SPADE    |4.5|16.62|
> |SNGAN-CoPE    |4.7|19.70|
>
> The number of parameters in the CoPE and ADD are very similar (the Hadamard product is replaced with the addition), while the 'SNGAN-CONC' includes additional parameters to account for the increased channels due to the concatenation. Interestingly, SNGAN-SPADE is faster than SNGAN; this speed improvement stems from conditional batch normalization (CBN). SNGAN samples in every layer from CBN, while SPADE only performs a Hadamard product.
>
> The second table below measures the number of parameters and the speed in the class-conditional generation experiment of sec.4.1 on CIFAR10.
>
> |    Method            |Parameters (million) |Speed (in ms)                         |
> |---------------------|-----------------|-----------------------|
> |GAN-CONC    |5.6|18.79|
> |GAN-ADD    |4.8|17.87|
> |SPADE        |3.9|9.96|
> |Π-net-SINCONC    |4.8|12.14|
> |Π-net        |4.8|19.12|
> |CoPE        |4.1|14.85|
>
> The results are similar to the ones reported for SNGAN. Notice that the Π-net-SINCONC and Π-net have similar number of parameters, but substantially different speed (over 50% overhead in Π-net). The difference is that Π-net uses CBN, while Π-net-SINCONC concatenates the classes in the input. Overall, the Π-net with CBN is the most costly in terms of speed.
>
> ________
>
> > What is the seed value we have taken for this experiment? Are those results are shown for one seed or an interval? If, interval what was the distribution that we sampled seeds?
>
> All our experiments are repeated five times, where the seed value is sampled from the framework’s (e.g., numpy) random seed generator.
>
> ________
>
> > Why does the setup work well in a GAN? Is there any theoretical evidence in the paper?
>
> Our main contribution is a general framework for the construction of networks of multiple inputs. This is not specific not only in the GAN setting, but it can be generalized to other generative frameworks, such as VAE in sec. 4.2. There is indeed evidence from the recent work of [1] that even second-order polynomials enlarge the hypothesis space of “standard” neural networks.
>
>
> [1] Jayakumar, Siddhant M., Wojciech M. Czarnecki, Jacob Menick, Jonathan Schwarz, Jack Rae, Simon Osindero, Yee Whye Teh, Tim Harley, and Razvan Pascanu. “Multiplicative Interactions and Where to Find Them.” In ICLR 2020.

---

### Official Review · Reviewer_41qE · 2021-07-19

**Rating:** 4
**Confidence:** 4

**Summary:**

This paper proposes CoPE, a new neural network architecture for conditional generative modeling. The architecture is derived from a coupled CP decomposition of the parameter tensors of multivariate polynomials, which gives rise to a recursive formula that can be interpreted as a neural network. The method is evaluated on a variety of conditional image generation tasks.


**Ethical Concerns:**

No ethical concerns.

**Limitations And Societal Impact:**

While limitations are discussed in Sect. 3.2, the proposed solutions to address the limitations have fundamental limitations of their own. I outline two below:

1) The proposed method for increasing the order of the polynomial in a parameter-efficient way (described on L156-165) assumes a specific factorization of the high-order polynomial. Whereas the original formulation can model any polynomial whose parameters are sufficiently low-rank tensors, the proposed method cannot.
2) The proposed method for improving stability of high-order polynomials (described on L166-175) only makes the outputs bounded, but does not make the gradients bounded, which means that a tiny change in the input can cause a huge change in the output. So I don't think this technique alone will overcome stability issues. More discussion should be included in the paper on how to get around this issue in practice.

**Main Review:**

Strengths:

The presentation is relatively clear (the supplementary material is especially so), and the method is evaluated on a broad range of tasks.
The proposed architecture is in the relatively under-explored area of polynomial neural networks, as opposed to an over-explored area.

Weaknesses:

The proposed architecture, CoPE, is a special case of \Pi-net, where the input variable is a concatenation of the conditioning input and the latent noise. This calls into question the novelty of the CoPE compared to \Pi-net. The main claim of novelty made in the paper (i.e. in Table 1 of the main paper) is that \Pi-net considers a single input variable only, whereas CoPE considers multiple input variables. But the two are equivalent once the multiple input variables are concatenated together. This is acknowledged on L555 in the supplementary material, and the claim is revised to \Pi-net being "not as flexible as the proposed CoPE" on L547 in the supplementary material. By flexibility, the authors refer to the fact that different structures can be imposed on different submatrices of the embedding matrices, \Lambda_[i]. I believe all the four differences between CoPE and \Pi-Net pointed out in Sect. E of the supplementary material boil down to this, and therefore, this forms the core methodological contribution made by this paper.

Based on a close reading of the supplementary material (in particular Sect. E), I understand the paper claims the following concrete contributions:
1) The precise way in which the embedding matrices are partitioned into submatrices (namely along the boundaries of the conditioning input and the latent noise)
2) The idea of sharing the columns of embedding matrices corresponding to the conditioning input across different orders (eqn. 19 of the supplementary material)
3) The idea of imposing convolution structure on the columns of embedding matrices corresponding to the conditioning input, but *not* on the columns corresponding to the latent noise
4) The concept of setting the columns of embedding matrices corresponding to the latent noise to zero for all orders above a threshold (though it's unclear if this concept was actually used)

I think there are a number of issues:

1) In terms of presentation, I would have preferred for these to be stated upfront in the main paper, rather than having them in a middle section of the supplementary and claiming the single variable input vs. multiple variable input distinction as the novel contribution in the main paper. The latter distinction comes down to the way the recursive formulas (Eqs. (2) and (17)) are written - the original novelty claim is essentially about the writing the formulas in terms of submatrices, rather than the full matrix. A new way of writing a formula is not a valid claim of novelty.
2) These four concrete contributions are more like design choices than methodological contributions. These normally don't constitute substantial contributions; however, because the paper is on a relatively under-explored area, it could be that these are actually non-trivial design choices and are important in obtaining good empirical results in practice. However, I don't see evidence of this: for example, in Figs. 10 and 17, there isn't a big qualitative difference between \Pi-Net and CoPE results. Furthermore, for several tasks (e.g.: super-resolution, edge-to-image), no qualitative comparison between \Pi-Net and CoPE results is shown. Moreover, no ablation study is provided for the different changes that are made relative to \Pi-Net. Given the limited methodological novelty, substantial improvements should be demonstrated over \Pi-Net to claim the non-trivialness of these design choices.

**Time Spent Reviewing:**

12 hours

---

> ### Author Response · Authors · 2021-08-09
> **Response to the reviewer 41qE**
>
> We appreciate the detailed comments of the reviewer and their elaborate review. We address the concerns below:
>
> > The main claim of novelty made in the paper (i.e. in Table 1 of the main paper) is that \Pi-net considers a single input variable only, whereas CoPE considers multiple input variables. But the two are equivalent once the multiple input variables are concatenated together.
>
> We agree with the reviewer that the two different formulations could learn a similar model. However, in practice concatenating two variables arising from diverse signals (e.g., noise and image) that potentially have vastly different dimensionality results in an underperforming model as highlighted by the performance of Π-net-SINCONC (tables 4-5). In this paper, we posit that different signals require different operations (e.g. convolutions versus concatenation) and verify this assumption in a wide range of tasks: there is a ~18% improvement on CIFAR10 class-conditional generation, and over 33% improvement on super-resolution.
> ________
>
>
> > The four concrete contributions are more like design choices than methodological contributions. These normally don't constitute substantial contributions.
>
>
> We agree with the reviewer that the implementation differences are closer to design choices and thus are stated in the supplementary. However, our main contribution is a flexible framework that allows for these design choices. In particular, using those choices, we derive two architectures (new architectures can be derived using different assumptions in the tensor factorizations).
>
> ________
>
> > Furthermore, for several tasks (e.g.: super-resolution, edge-to-image), no qualitative comparison between \Pi-Net and CoPE results is shown.
>
> In some cases, extending directly Π-net was not trivial; for instance in super-resolution we would need a single layer that can both operate on the low-resolution image and the vector-based noise. To our knowledge, two popular strategies would be:
>
> * Vectorize the low-resolution image, concatenate it with the noise, and use a fully-connected layer. However, this would lose all the spatial correlations of the low-resolution image.
>
> * Convert the noise into the dimensions of the low-resolution image. However, that imposes spatial correlations on the noise which are unwanted.
>
> As we mention in the paper, the inductive bias of the network is a critical component for the success of the method and we demonstrate how CoPE can be successful in tasks like super-resolution and other dense regression tasks.
>
> ________
>
> > The proposed method for increasing the order of the polynomial in a parameter-efficient way (described on L156-165) assumes a specific factorization of the high-order polynomial. Whereas the original formulation can model any polynomial whose parameters are sufficiently low-rank tensors, the proposed method cannot.
>
> We respectfully disagree with the reviewer. As the reviewer noted above, the presented model uses similar tensor factorizations as Π-net and therefore it inherits the low-rank property of the decomposition. Using equation 2, you can build a polynomial of arbitrary degree in a parameter-efficient way due to the low-rank CP decomposition as well as the sharing (e.g., equation 19). However, we discover that increasing the degree of the polynomial by using a product of polynomials performs better experimentally. We will clarify this design choice in the revised version of the manuscript.
> ________
>
> > The proposed method for improving stability of high-order polynomials (described on L166-175) only makes the outputs bounded, but does not make the gradients bounded, which means that a tiny change in the input can cause a huge change in the output. So I don't think this technique alone will overcome stability issues. More discussion should be included in the paper on how to get around this issue in practice.
>
> We agree that the gradients are not guaranteed to be bounded. However, this was not an issue in practice in our conditional generation experiments. The issue of instability has been previously reported (e.g., in Π-nets). Dealing with this issue is a significant task that is however beyond the scope of this paper.

---

> > ### Comment · Reviewer_41qE · 2021-08-27
> > **Thanks for the response**
> >
> > I appreciate the authors' response. If I understand correctly, the main claim of novelty is that in CoPE, different layers can be applied to the input image and the noise (for example, a convolution layer to the image and a fully-connected layer to the noise), whereas in Π-net, the same layer needs to be applied to the concatenation of the input image and the noise. This latter interpretation of Π-net is too narrow in my opinion - the first layer in Π-net can be any linear operator and does not need to be either a fully-connected layer or a convolution layer. Applying different linear operators to different components of the input simply corresponds to having a block-diagonal structure in the weight matrix. So, essentially the claim of novelty is the imposition of such a structure, which I don't believe to be substantial.
> >
> > Thanks for pointing out the comparison to Π-net-SINCONC. While CoPE does do a lot better than Π-net-SINCONC on CIFAR10 class-conditional generation (FID of 34.35 compared to 71.81), its improvement relative to Π-net is much more modest (34.35 compared to 37.26). On super-resolution (Table 5), CoPE is only compared to Π-net-SINCONC, and not Π-net. I have two follow-up questions:
> >
> > 1. Since Π-net does better than Π-net-SINCONC, I assume Π-net is a stronger baseline and should be the yardstick for measuring performance improvements. Since the authors focus on comparisons to Π-net-SINCONC, is there any reason to prefer Π-net-SINCONC as the baseline?
> > 2. The authors state that the reason for not including a comparison to Π-net on super-resolution (Table 5) is because extending it to the problem setting is not trivial because a single layer must be applied to the concatenated input. However, a comparison to Π-net-SINCONC is included for super-resolution, and according to L734-739 in the supplementary material, the only differences with Π-net are (1) conditional batch normalization vs. regular batch normalization and (2) concatenation of the input with the noise in Π-net-SINCONC. This concatenation is what was claimed to be not trivial to do in the rebuttal, but since Π-net-SINCONC is included as a baseline, I don't understand why doing the same thing in the context of Π-net would be non-trivial.
> >
> > For the concern I raised regarding the proposed method of building high-order polynomials by composing together polynomials (described on L166-175), while the authors state their disagreement respectfully, the rebuttal doesn't actually invalidate the point I raised. Suppose one has a rank-1 order-n polynomial - the number of parameters in this polynomial is nd, where d is the input dimension. Using the proposed method, we would have two rank-1 polynomials with orders l and m, such that lm = n. The sum of the number of parameters in these polynomials is (l+m)d. There is no way that *any* polynomial with nd = lmd parameters can be represented exactly with (l+m)d parameters. So it can only work with a special class of rank-1 order-n polynomial that admits a particular factorization. The rebuttal states that one can build a high-order polynomial using the proposed method, which I agree with. What I pointed out in my review is that there are some high-order polynomials that *cannot* be represented using the proposed method.
> >
> > Regarding instability, since a way to reduce it was claimed as a contribution, I would think that dealing with it falls within the scope of the paper. So I find it somewhat contradictory that the rebuttal claims that it's not an issue in practice.

---

> > > ### Author Response · Authors · 2021-08-29
> > > **Response to the reviewer**
> > >
> > > We are thankful to the reviewer 41qE for their additional comments. We respond below to the comments.
> > >
> > > > Since Π-net does better than Π-net-SINCONC, I assume Π-net is a stronger baseline.
> > >
> > >
> > >
> > > We agree with the reviewer, that Π-net is a stronger baseline in class-conditional generation experiments. The reason is that the authors are using Conditional batch normalization (please check implementation details of the official Π-net implementation [here](https://github.com/grigorisg9gr/polynomial_nets/blob/master/image_generation_chainer/gen_models/cnn_gen_custom_prodpoly.py#L170)). However, this conditional batch normalization effectively applies a batch normalization layer per class (i.e. separate statistics per discrete class). Experimentally, learning a different batch normalization layer for each class works well, but it relies on discrete classes.
> > >
> > >
> > >
> > > This is the reason that we have not included Π-net as a baseline in the super-resolution experiment since the conditioning variable (i.e. low resolution image) has continuous values. Specifically, we have followed the official implementation in Π-net baseline (with conditional batch normalization that works for discrete classes) and enabled the concatenation with the Π-net-SINCONC baseline (which does NOT have conditional batch normalization, but regular batch normalization). We will clarify this in the camera-ready version of the paper.
> > >
> > > ___________________
> > >
> > >
> > >
> > > > The rebuttal states that one can build a high-order polynomial using the proposed method, which I agree with. What I pointed out in my review is that there are some high-order polynomials that cannot be represented using the proposed method.
> > >
> > >
> > >
> > > We agree with the reviewer; not all polynomials can be exactly reconstructed using less parameters than the source polynomial. However, we can uniformly approximate any source polynomial as closely as desired (Stone-Weierstrass Theorem).
> > >
> > > __________
> > >
> > >
> > >
> > > > Regarding instability, since a way to reduce it was claimed as a contribution, I would think that dealing with it falls within the scope of the paper. So I find it somewhat contradictory that the rebuttal claims that it's not an issue in practice.
> > >
> > >
> > >
> > > We mention what we observed in practice, i.e., that using normalization schemes did not cause any issues in learning the polynomials. We believe this is a useful observation for practitioners interested in studying or extending our method. We agree with the reviewer that the stability or learning dynamics are significant topics, but those are separate topics that we intend to study this in the future.

---

### Decision · Program_Chairs · 2021-09-28

**Decision:**

Accept (Poster)

**Comment:**

After discussion, there's general consensus among reviewers to reject the work, and with no one strongly in favor of acceptance. The paper's contributions could be better highlighted against related work such as Pi-Nets, and experimental Results are not thorough (both criticisms shared across reviewers). I highly recommend the authors use the reviewer feedback to improve their work in any future conference submission.

**Consistency Experiment:**

NeurIPS has a long history of experimentation. In 2014, NeurIPS ran an experiment in which 10% of submissions were reviewed by two independent committees to quantify the randomness in the review process. This year, we repeated a variant of this experiment to see how the quality of the review process has changed over time.  This paper was part of the experiment and was therefore assigned to two committees (consisting of reviewers, an Area Chair, and a Senior Area Chair) that reached independent decisions.  If both committees made the same recommendation, this recommendation was followed. If a single committee recommended acceptance, the paper was accepted (with the exception of a few cases in which the other committee identified what we considered a fatal flaw, e.g., an error in a key result).

This copy’s committee reached the following decision: **Reject**

The other committee assigned to the paper recommended **Accept (Spotlight)**.  You can find the other set of reviews, along with any follow up discussion with the authors here:
https://openreview.net/forum?id=wPA_5Wsjt8i